# Mobilisation thresholds for coral rubble and consequences for windows of reef recovery

Tania M. Kenyon[1]*, Daniel Harris[2], Tom Baldock[3], David Callaghan[3], Christopher Doropoulos[4], Gregory Webb[2], Steven P. Newman[5] and Peter J. Mumby[1].

[1]Marine Spatial Ecology Lab, School of Biological Sciences, The University of Queensland, St. Lucia, Australia.

[2]School of Earth and Environmental Sciences, The University of Queensland, St. Lucia, Australia.

[3]School of Civil Engineering, The University of Queensland, St. Lucia, Australia.

[4]Commonwealth Scientific and Industrial Research Organisation, St. Lucia, Australia.

[5]Banyan Tree Marine Laboratory, Vabbinfaru, North Male' Atoll, Maldives.

*Correspondence to:* Tania M. Kenyon (tania.kenyon@uq.net.au)

Keywords: coral reef; hydrodynamics; sediment transport; rubble stabilisation; Maldives; Vabbinfaru; disturbance.

**Abstract**

The proportional cover of rubble on reefs is predicted to increase as disturbances increase in intensity and frequency. Unstable rubble can kill coral recruits and impair binding processes that transform rubble into a stable substrate for coral recruitment. A clearer understanding of the mechanisms of inhibited coral recovery on rubble requires characterisation of the hydrodynamic conditions that trigger rubble mobilisation. Here, we investigated rubble mobilisation under regular wave conditions in a wave flume and irregular wave conditions *in-situ* on a coral reef in the Maldives. We examined how changes in near-bed wave orbital velocity influenced the likelihood of rubble motion (e.g., rocking) and transport (by walking, sliding or flipping). Rubble mobilisation was considered as a function of rubble length, branchiness (branched vs. unbranched), and underlying substrate (rubble vs. sand). The effect of near-bed wave orbital velocity on rubble mobilisation was comparable between flume and reef observations. As near-bed wave orbital velocity increased, rubble was more likely to rock, be transported and travel greater distances. Averaged across length, branchiness and substrate, loose rubble had a 50% chance of transport when near-bed wave orbital velocities reached 0.30 m/s in both the wave flume and on the reef. However, small and/or unbranched rubble pieces were generally mobilised more and at lower velocities than larger, branched rubble. Rubble also travelled further distances (~2 cm) on substrates composed of sand than rubble. Importantly, if rubble was interlocked, it was very unlikely to move (<7% chance) even at the highest velocity tested (0.4 m/s). Furthermore, the probability of rubble transport declined over 3-day deployments in the field, suggesting rubble had snagged or settled into more hydrodynamically-stable positions within the first days of deployment. We expect that snagged or settled rubble is transported more commonly in locations with higher energy events and more variable wave environments. At our field site in the Maldives, we expect recovery windows for binding (when rubble is stable) to predominantly occur during the calmer north-eastern monsoon when wave energy impacting the atoll is less and wave heights are smaller. Our results show that rubble beds comprised of small rubble pieces and/or pieces with fewer branches are more likely to have shorter windows of recovery (stability) between mobilisation events, and thus be good candidates for rubble stabilisation interventions to enhance coral recruitment and binding.

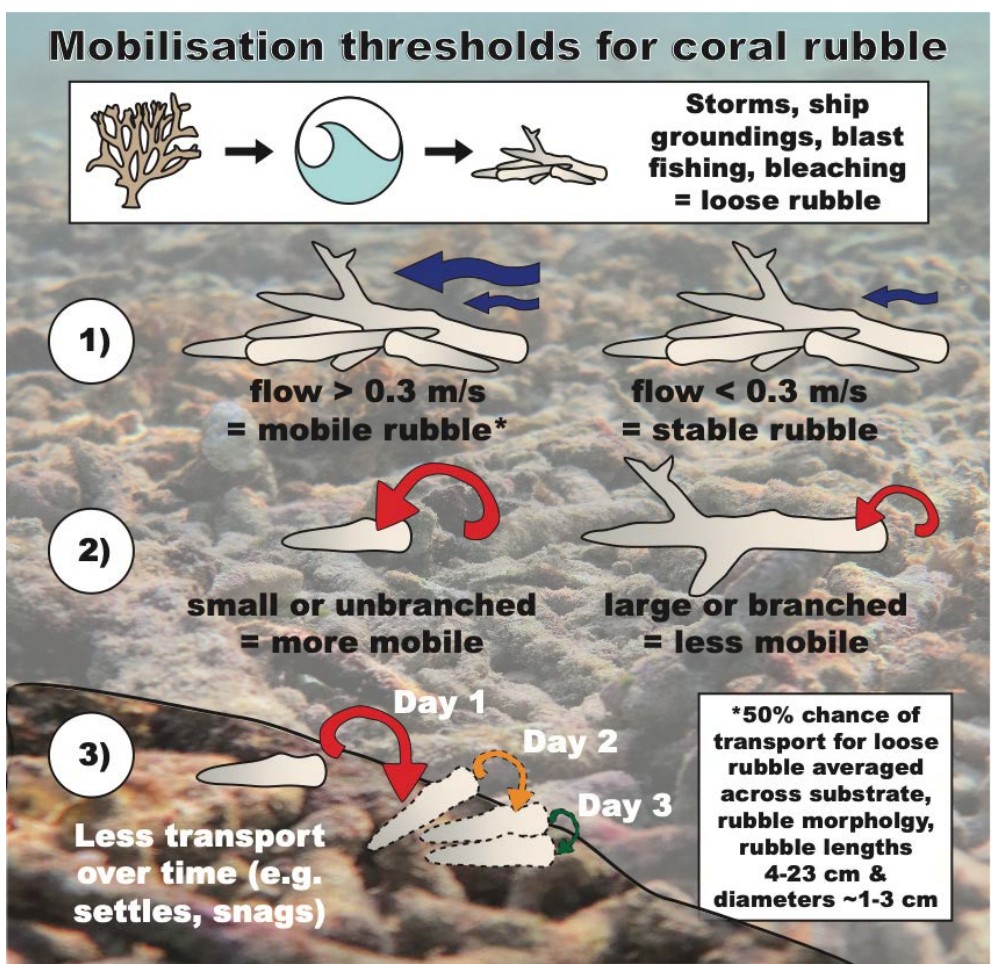


## 1 Introduction

Coral reefs routinely experience disturbances that physically break up reef rock and live coral skeletons into fragments within the cycle of erosion and accretion (Scoffin 1992, 1993; Blanchon and Jones 1997; Blanchon et
al. 1997). Some of these coral fragments reattach, contributing to asexual recruitment (Highsmith, 1982) while others die and contribute to the accumulation of rubble on the substrate, which is naturally high on some reefs
(Thornborough, 2012; Davies, 1983). Disturbances, including storms, dynamite fishing, ship groundings and trampling, can cause large accumulations of rubble (Fox and Caldwell, 2006; Viehman et al., 2018; Gittings et
al., 1994; Hawkins and Roberts, 1993; Scoffin, 1993; Woodley et al., 1981a). Coral bleaching and disease do not directly reduce structural complexity, but result in *in-situ* mortality and eventual breakdown of the coral skeleton
into rubble (Scoffin and McLean, 1978; Aronson and Precht, 1997). As sea surface temperatures rise, storm and cyclone intensity is predicted to increase, particularly in the Atlantic and West Pacific (Meehl et al., 2007; Knutson
et al., 2010), and bleaching events are becoming more frequent (Hughes et al., 2018; Hoegh-Guldberg, 1999). Reefs are predicted to 'flatten' into systems with high rubble:coral ratios over time as recovery windows between
disturbance events become increasingly smaller (Lewis, 2002; Hoegh-Guldberg et al., 2007; Alvarez-Filip et al., 2009). High rubble cover can persist in an unstable state for years to decades on some damaged reefs (Dollar and
Tribble 1993, Lasagna et al. 2008, Chong-Seng et al. 2014, Viehman et al. 2018, Fox et al. 2019) and can also

form persistent rubble beds that remain for centuries to millennia (Liu et al., 2016; Clark et al., 2017; Yu et al., 2012; Montaggioni, 2005).

A key determinant of recovery on reefs where large tracts of coral have been turned to rubble is the stability of rubble. Rubble mobilisation correlates with flow velocity (Cheroske et al., 2000; Bruno, 1998; Viehman et al., 2018), wind speed and wave energy (Cameron et al. 2016), and in meso-tidal regions with water depth, inundation duration and tidal phase (Thornborough 2012). Hydrodynamic forcing above a certain threshold will cause rubble to be mobilised by sliding or flipping (Viehman et al., 2018). Moreover, the loss of structurally-complex framework reduces a coral reef's capacity to dissipate hydrodynamic energy, leading to greater near-bed orbital flow velocities over rubble beds (Guihen et al., 2013). Frequent mobilisation events in a rubble bed can hinder the recovery of coral assemblages by increasing mortality of sexual and asexual coral recruits within the rubble bed through abrasion and smothering (Clark and Edwards, 1995; Brown and Dunne, 1988; Kenyon et al., 2020). Furthermore, mobilisation could break binds formed by encrusting organisms between individual rubble pieces, preventing the binding of rubble into a stable substrate (Rasser and Riegl, 2002). Rubble mobilisation under everyday wave conditions (as opposed to storm events) has resulted in a lack of recovery of coral assemblages over a period of 6 (Viehman, 2017) to 17 years (Fox et al., 2019) post-disturbance. Under future climate scenarios, sea level rise might also result in enhanced rubble mobilisation (Kenyon et al., 2023) via increased wave orbital velocities on some reefs (Baldock et al., 2014a, b). Implications of the persistence of rubble beds with low structural complexity extend beyond reduced coral cover, including reduced fish abundance, diversity and fisheries productivity (Rogers et al., 2018; Luckhurst and Luckhurst, 1978; Graham et al., 2006) and reduced coastal protection (Harris et al., 2018a; Ferrario et al., 2014). To predict and manage the recovery potential of post-disturbance rubble beds, we must understand the drivers and frequency of rubble mobilisation.

Although disturbances attributed to hydrological regimes are well studied in some systems, e.g., substrate stability in streams and intertidal areas (Townsend et al., 1997; Suren and Duncan, 1999; Hardison and Layzer, 2001; Sousa, 1979), studies on rubble mobilisation on coral reefs are in their infancy. Sediment transport studies commonly deal with smaller particles than rubble, including sand, silt and clay (<2 mm according to the modified Udden-Wentworth grain-size scale) (Blair and McPherson, 1999). As hydrodynamic energy increases, sediment from a larger range of size classes are transported (Komar and MIller, 1973; Kench, 1998b; Nielsen and Callaghan, 2003), in some cases on vast scales during cyclones and hurricanes (Keen et al., 2004; Hubbard, 1992). Attention has also been given to movement initiation of boulders from 20 kg to ~290 t (Nott, 2003, 1997; Nandasena et al., 2011; Etienne and Paris, 2010; Imamura et al., 2008; Kain et al., 2012). While coral rubble can be boulder-sized (Rasser and Riegl, 2002), clasts are typically much smaller, averaging 5–30 cm in length and as small as 1 cm (Highsmith et al., 1980; Fong and Lirman, 1995; Heyward and Collins, 1985; Dollar and Tribble, 1993; Kay and Liddle, 1989). Few studies have monitored mobilisation of rubble in this size range with knowledge of the wave environment and flow rate estimates, particularly in field environments (Cheroske et al., 2000; Viehman et al., 2018).

The probability that rubble will remain stable depends not only on hydrodynamic forcing but also on rubble characteristics (e.g., size and shape), and the type and bathymetry of the underlying substrate (the 'pre-transport environment') (Nandasena et al., 2011; Nott, 2003). While their densities may vary slightly, research on the survivorship of live coral fragments provides insight into the behaviour of dead rubble pieces. Studies show that

the likelihood of coral fragment survival decreases with decreasing size (Smith and Hughes, 1999), likely due to increased mobilisation of smaller fragments (Hughes, 1999). Fragments with non-branching morphologies have reduced survival compared to those with branching morphologies (Tunnicliffe, 1981; Heyward and Collins, 1985; Smith and Hughes, 1999), likely due to greater mobility and increased smothering of less complex shapes. The stability and survival of fragments also varies with substrate type and bathymetry. Live fragments tend to survive more commonly on rubble than on sand substrates (Heyward and Collins, 1985; Bruno, 1998; Bowden-Kerby, 2001; Prosper, 2005; Kenyon et al., 2020) and are transported further in reef slope zones where gravity assists mobilisation, than in planar lagoons with low slope angles (Smith and Hughes, 1999). Steep slopes can foster downslope transport and the formation of a rubble talus (Rasser and Riegl, 2002; Dollar and Tribble, 1993). Rubble beds on reef slopes generated by intense disturbances and comprising small, unbranched rubble, are therefore likely at high risk of mobilisation. However, to our knowledge there has been no study where the threshold of mobilisation for individual rubble pieces of varying shapes and sizes, and on different substrate types and slopes, has been empirically determined in both controlled and field settings.

Here, we report how the probability of rubble mobilisation changes as near-bed wave orbital velocity increases under average (everyday) hydrodynamic conditions. We quantified the thresholds required to mobilise coral rubble, and identified effects of rubble size and morphology, underlying substrate type, and slope angle, on the likelihood of mobilisation. Experiments were conducted in a controlled, wave flume environment, and replicated as closely as possible in the field to extend findings from a regular (monochromatic) wave environment to an irregular wave environment. We hypothesised that the probability of rubble mobilisation would decrease as: (i) rubble size increases; (ii) morphological complexity increases (of both the rubble and of the substrate type); and (iii) as the slope angle decreases (and the contribution of gravity subsequently decreases). Managers of reefs that exhibit a significant increase in rubble cover can use the mobilisation estimates reported here, coupled with knowledge of the reef's hydrodynamic exposure (e.g., publicly accessible wind data, wave climate estimates), rubble typology, and other environmental factors, to predict the frequency of everyday rubble mobilisation and the likelihood of natural rubble stabilisation and recovery.

## 2    Methods

### 2.1    Mobilisation in flume

To determine the velocity required to mobilise rubble, trials were conducted in a wave flume (l: 20 m; w: 2 m; d: 1.2 m) using a DHI Technologies piston wave maker (Figure 1 a-b; see Baldock et al. 2017 for general description). Cylindrical rubble pieces (from branching coral species) were collected from Lizard Island, Great Barrier Reef in 2017 after the 2016 bleaching event. Rubble was divided into four size categories based on axial length (4–8 cm; 9–15 cm; 16–23 cm; and 24–36 cm; all with a diameter of 1–2 cm) and two 'branchiness' categories: unbranched (if rubble had no branches > 1 cm length) and branched (if rubble had branches > 1 cm length), with 5–10 pieces in each size/branchiness group. The size range of rubble used in the laboratory phase of the study is consistent with that commonly observed on reefs following natural and anthropogenic disturbances (Highsmith et al., 1980; Fong and Lirman, 1995; Heyward and Collins, 1985; Dollar and Tribble, 1993), as well

as the size range (1 – 27 cm, mean 7 cm) of 440 rubble pieces measured from Vabbinfaru Reef (which also suffered bleaching in 2016) where the field portion of this study was undertaken.
The mobilisation of 'loose' (not interlocked) cylindrical rubble was tested on two substrate types: sand and rubble. Beach sand ~2 cm (grain size $d_{50}$=0.28mm) deep was spread over the flume base to form the sand substrate (Figure
1 a). The rubble substrate comprised 'Serenity Aquatics' Coral Rubble (l: 3–5 cm) glued to a plywood base (l: 2 m; w: 1 m) that lay on the concrete base of the flume (Figure 1 b). The mobilisation of interlocked rubble was
tested on a second rubble substrate, which comprised a stainless-steel mesh with rubble of mean length 9 cm (3– 20 cm range) attached with cable ties (Figure S1). The height of both bases averaged 2 cm, although some rubble
pieces protruded up to 5.5 cm in the second base. Small and medium-sized branched cylindrical rubble of 4–15 cm length were manually interlocked with the second rubble base prior to testing. Larger rubble and unbranched
rubble could not be suitably interlocked and therefore were not tested on the second rubble base.

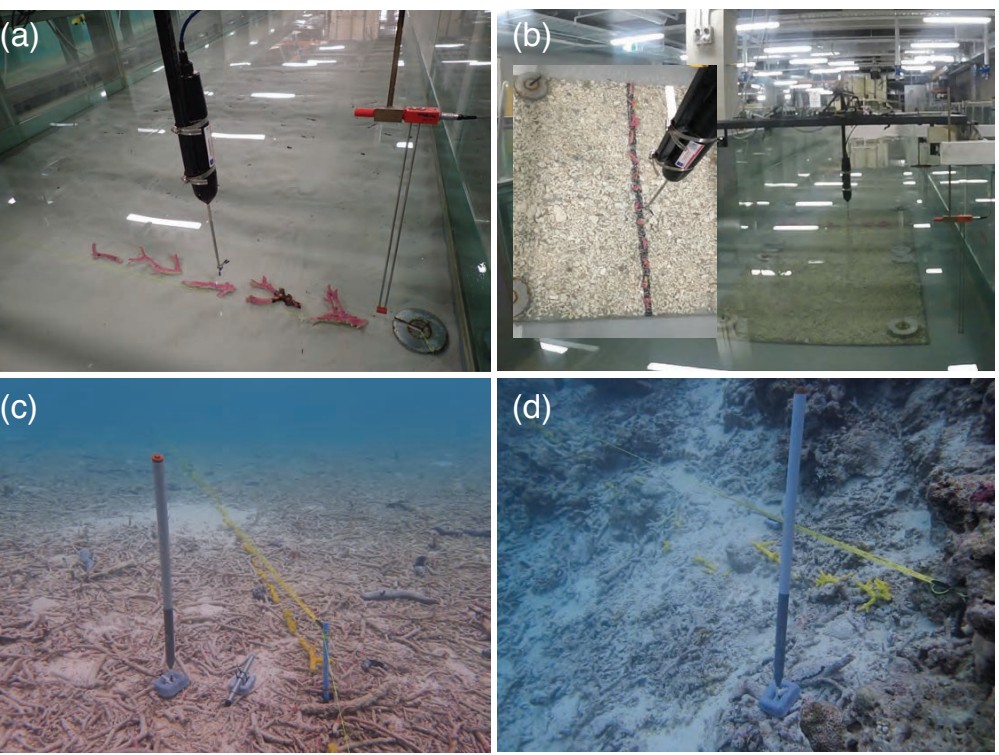

**Figure 1: Experimental rubble (painted) lined up along a reference line (a) in flume with sand substrate; (b) in flume with rubble substrate to test loose pieces, and inset close-up view; (c) in the field in a shallow reef flat site (2–3 m); and**
**(d) in the field in an exposed deep site (6–7 m, western reef) (Source: T Kenyon).**

Rubble was placed along a reference line parallel with the wave paddle, with the long axis normal (perpendicular)

to flow to identify the minimum velocity threshold (short-axis normal to flow requires a higher threshold) (Figure 1 a-b). The wave maker ran 30-second bursts of regular (monochromatic) waves, starting at water depth ($h$) =
0.42 m, wave height (H) = 0.05 m and wave period (T) = 1 s. Wave height (H) was increased in 0.02 m increments at the same period (T). Three replicate waves were run for each wave height and period combination and the
movement type for each rubble piece was recorded for each run. Binding could be prevented and weak binds damaged by even small rocking motions, and corals could be abraded and smothered by rubble transport and

flipping. Thus, the movement categories chosen were: no movement (rubble remained stable and in the same position); rocking (rubble rocked back and forth and in some cases rotated, but remained in the same position); transport by walking/sliding (rubble walked/skittered or slid away from initial position); and transport by flipping (rubble overturned at least once). If a piece rocked, then slid and then flipped within one run, the movement type was marked as flipping, because more force is required to overturn a piece than to rock or slide it (Viehman et al., 2018; Imamura et al., 2008). The near-bed wave orbital velocity (m/s) for each run was estimated using the Soulsby Cosine Approximation (Soulsby, 2006), shown to produce similar estimates to linear wave theory (within 0.01 m/s) (Figure S2, Table S1).

To determine whether scaling effects were necessary to compare velocity thresholds between flume and field conditions, we derived a relationship for the contribution of the inertia force to the total maximum force as a proportion of the drag force, for all wave conditions tested. Total force depends on both the inertia force and drag force components, and while the inertia component is dependent on velocity and wave period, the drag component is solely dependant on velocity (Table S1). Thus, where conditions are determined to be drag dominated, rubble movement depends primarily on velocity, and valid comparisons between flume and field can be made despite their variance in wave period. The contribution of inertia force to the total maximum force for each wave height and period combination in the flume, based on an average coral diameter of 1.64 cm (range ~1-2 cm), is shown in Table S1. Only 19 out of 71 wave conditions in the flume have the potential for the inertia force to be significant, and of those, only 7 had a $\frac{F_I}{F_D}$ ratio >2, meaning that nearly all wave conditions in the flume led to drag-dominated conditions. Furthermore, the inertial component decreases as velocity increases (Figure S5), and inertial forces were negligible at the 50% and 90% transport thresholds (see results for further explanation). The flume and field experiments are therefore comparable without scaling effects.

## 2.2 Mobilisation in field

To compare flume trials to a natural reef setting, trials were conducted in the field across different reef zones on Vabbinfaru Reef, North Male' Atoll, Maldives (4°18′35″N, 73°25′26″ E). The reef crest is 0.6–1.5 m below mean sea level and surrounds a shallow subtidal reef flat (~1.17 m below mean sea level) and sand cay (Morgan and Kench, 2012). Tidal ranges in the region are microtidal: 0.6 m and 1.2 m during neap and spring tides, respectively (Kench et al., 2009). When this study was conducted, rubble cover was high on the reef flat and on the reef slope following bleaching events in 1998 and 2016 (Zahir et al., 2009; Perry and Morgan, 2017). Coral cover on the reef crest was reduced from 50–75% down to 9% (Banyan Tree Marine Laboratory, unpublished data). The Maldives has two distinct monsoon seasons: the wet from April to October during which stronger winds blow predominantly from the southwest; and the dry from November to March where north-eastern winds are gentler on average (Kench et al. 2006, Figure S3). The north-eastern and western monsoons correspond to minimum and maximum incident ocean swell conditions, respectively (Kench et al. 2009). Daily winds at Vabbinfaru average 10 knots (mean daily maximum 19.8 knots) and are predominantly westerly, while the southeast region of the reef is relatively sheltered year-round (Figure 2 b) (Beetham & Kench 2014).

Previous studies on Vabbinfaru reef suggest that sediment transport is largely controlled by wind-driven waves associated with the western monsoon, rather than tidally-driven currents (Morgan and Kench, 2014a). Thus,

rubble mobilisation was related to near-bed wave orbital velocity. To capture a gradient in wave energy, rubble mobilisation was tracked in different sites and monsoon seasons. Fifteen field sites were delineated across reef flat (~2 m depth), shallow reef slope (2–3 m) and deeper reef slope (6–7 m) environments on the sheltered (southeast) and comparatively exposed (western) sides of the island (Figure 2 a). The field trials were conducted in all sites in the north-eastern monsoon (late November 2017 to January 2018) and again in the western monsoon (early August to September 2018).

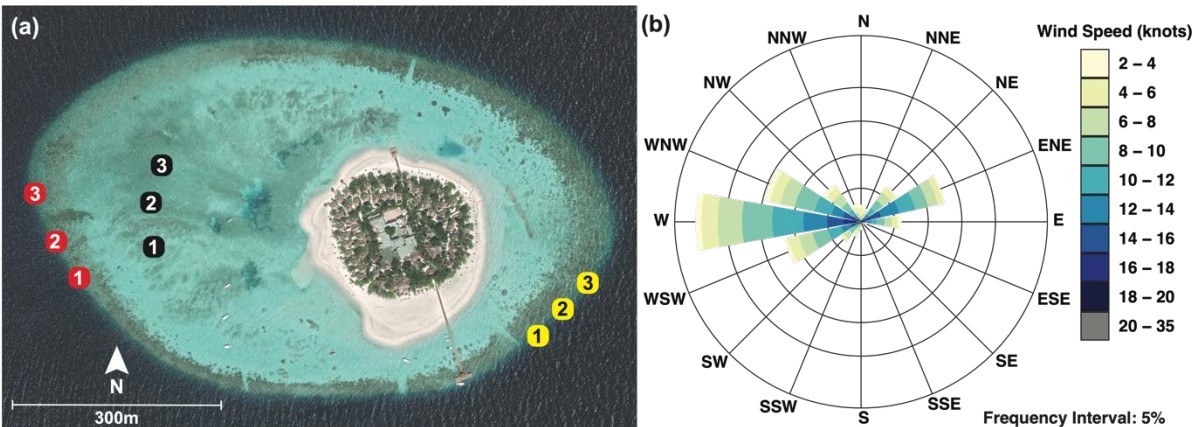

**Figure 2: (a) Field sites at Vabbinfaru platform: Three 2–3 m sites on the reef flat (black); three site locations on the exposed western reef slope (red), each comprising a shallow (2–3 m) and deep (6–7 m) site; and 3 site locations on the sheltered southeast reef slope (yellow), each comprising a shallow and deep site (Source: © Google Earth). (b) Windrose of mean wind speed (knots) and wind direction data measured at nearby Hulhumale ranging 1985-2018 for both seasons (Data source: Maldives Meteorological Service, Government of Maldives).**

The wave environment in each of these sites and seasons was characterised using INW Aquistar ® PT2X 30 psia pressure loggers placed on the seabed and recording continuously at 2 $Hz$ (Figure 1 c). Using known processing methods (Harris et al., 2018b, 2015), records from the pressure loggers were low-pass filtered to remove instrument noise and high-pass filtered to remove infragravity effects (at 0.05 $Hz$), then split into 30-minute runs to remove tidal influence (Hughes and Moseley, 2007). Pressure was converted to depth, and wave spectra for each 30-minute run were calculated between 0.0033-0.33 $Hz$ using the Welch method for computing power spectral densities from 3600 sample records, to obtain significant wave height ($H_s$) and peak wave period ($T_p$). The near-bed wave orbital velocity (U) was then estimated for each 30-minute run using linear wave theory using Eq. (3).

(3) $U = \frac{H_s}{2 \sinh(kh)} \cdot \frac{2\pi}{T_p}$       where the wave number ($k$) was determined by solving Eq. (4)

(4) $\varpi^2 = gk \sinh(kh)$       where ω is the wave radian frequency ($2\pi/T_p$), $h$ is water depth, and g the acceleration due to gravity.

The contribution of the inertia force to the total maximum force as a proportion of the drag force was estimated for each $H_s$ and $T_p$ combination used in the field analysis, based on an average coral diameter of 1.69 cm (range ~1-3 cm) (Table S2). Only 1 out of 90 wave conditions in the field had the potential for inertia to be significant,

meaning that most conditions in the field were drag-dominated. Furthermore, this one condition corresponded to a very low velocity (0.016 m/s), far from the reported 50% and 90% transport threshold velocities.
Rubble movement was tracked while the wave environment was measured, to correlate rubble mobilisation with near-bed wave orbital velocity. At each site and in each season, ~20 marked (painted yellow) rubble pieces of
axial length category 4–8 cm, ~20 pieces 9–15 cm and ~10 pieces 16–23 cm of both branched and unbranched varieties were placed along and directly beneath a reference string strung parallel to the reef crest (Figure 1 c-d).
A black dot was painted on the underside of each piece. The substrate beneath the rubble was recorded as either sand, rubble or hard carbonate, and the slope angle was measured at 50-cm intervals along the reference string
using a spirit level and right-angle set square. As the depth on the reef slope likely excluded swash effects, the net direction of mobilisation was expected to be downslope aided by gravity, rather than upslope with wave direction.
Mobilisation direction on the reef flat, however, was expected to be shoreward. Generally, reef flat sites were characterised by flatter slopes, shallow reef slope sites by gentle slopes, and deeper reef slope sites by steeper
slopes (Figure 1 c-d). The perpendicular distance from the reference string to each rubble piece was recorded over three days, approximately 24, 48 and 72 hours after deployment. A transect tape was laid along the reference
string to also record the point along the tape with which the rubble piece aligned. These two measurements were used to calculate the diagonal distance travelled by the rubble piece during each 24-hour interval over three days.
Whether or not the piece rotated or flipped was also recorded (if ≥ 50% of the black dot was visible). A piece was only considered to have moved if it was > 1 cm from its starting point. This buffer provided a degree of
conservatism to account for possible variations in the angle of gaze looking down on the reference string. Rocking movements could not be recorded *in-situ* as rubble pieces were not continually observed.
From the 30-minute runs across each 3-day period and site (144 each period and site), the fastest wave orbital velocity (calculated from significant wave height and peak wave period) was selected for each day, to regress
with observed rubble movement on that day. A total of the 90 fastest wave orbital velocities were thus used in the analyses that included all three days (1 velocity per day x 3 days x 15 sites x 2 seasons), and 30 were used in the
analyses that included the first day only (1 velocity for each 'day 1' x 15 sites x 2 seasons).

### 2.3    Statistical analyses

The movement categories of rocking, transport, and flipping (in the flume), and transport and flipping (in the field), were modelled as binary (Bernouli) responses, and classed as either a '0' or a '1' depending on the analysis
(Table 1). For example, when modelling the probably of transport in the flume, rubble was classed as '0' if it did not move or rocked only, and '1' if it walked/slid or flipped. Movements of walking, sliding and flipping were
considered in this case in order to compare mobilisation thresholds across flume and field (transported rubble in the field could have moved by any of these three movement types) (Table 1). Similarly, when modelling the
probably of flipping in the flume, rubble was classed as '0' if it did not move, rocked, walked/slid, and as a '1' only if it flipped. All analyses were conducted in R (R Core Team, 2020). For all models, backwards step-wise
selection was used to remove non-significant terms, whereby reduced models were compared to full models using the corrected Akaike Information Criterion (AICc) with package "MuMIn" (Bartoń, 2020). Model assumptions
were assessed using diagnostic plots.

**Table 1 Rubble movement types associated with each type of analysis from flume observations (i.e., probability of rocking, transport and flipping for loose, not interlocked, cylindrical rubble) and the analysis from field observations to which each was compared.**

| Flume analyses | Movement types classed as '0' | Movement types classed as '1' | Comparison to which field analyses |
|---|---|---|---|
| Rocking | No movement | Rocking (all other movement types excluded for this analysis) | N/A (Rocking could not be distinguished in the field as rubble was not observed continuously). |
| Transport | Rocking; or No movement | Walking/sliding; or Flipping | Transport >1 cm |
| Flipping | Walking/sliding; Rocking; or No movement | Flipping | Flipping |

The probability of flipping alone may have been underestimated in the field, i.e., a rubble piece might have rolled a complete 360°, meaning the black dot was again on the underside and not visible at the time of observation. Thus, the most appropriate comparison of mobilisation thresholds in the flume and field was between the threshold of transport in the flume for loose (not interlocked) cylindrical rubble and the threshold of transport in the field.

### 2.3.1    Mobilisation in flume

To identify the effects of rubble and substrate characteristics on the mobilisation of loose (not interlocked) rubble, logistic regression models (glm) were run using the base R 'stats' package, with the type of movement as the response variable and velocity, rubble size, branchiness, substrate and all interaction terms up to $3^{rd}$ order interactions, as explanatory variables. The analysis of the probability of rocking only considered trials where rocking (no transport) was the greatest movement observed. Interactions were investigated by conducting pairwise comparisons across levels of factors at velocities of 0.1 m/s, 0.2 m/s, 0.3 m/s and 0.4 m/s using the 'emmeans' package with Tukey adjustment (Lenth, 2020). It is expected that rubble beds *in situ* contain a variety of shapes and sizes of pieces and span multiple substrate types. Thus, to determine the threshold velocities at which 50% and 90% of rubble are transported, averaged across all rubble sizes, shapes and substrates, a reduced model was run with the type of movement as the response variable and 'velocity' as the sole explanatory variable. This model only used data for rubble of lengths ranging 4–23 cm (no 24–39 cm size class), to be consistent with the range of rubble used in the field and thus make thresholds comparable.

The mobilisation of interlocked rubble was analysed separately, and logistic regression models included 'any movement' (movement types were combined due to low mobilisation observations) as the response variable and velocity, rubble size and a velocity:size interaction as explanatory variables. To determine the most common movement types for interlocked rubble, another model was run using 'any movement' as the response variable, velocity, rubble size, movement type and interactions as explanatory variables (although only movement type remained in the model).

### 2.3.2 Mobilisation in field

To firsly characterise near-bed wave orbital velocities for each habitat and season, the package 'glmmTMB' (Brooks et al., 2017) was used to fit a mixed-effects model with a gamma distribution, with the fastest near-bed wave orbital velocity (m/s) as the response variable. Due to the lack of deep sites on the reef flat, leading to an unbalanced design, aspect and depth were combined to form a new variable 'habitat'. Habitat was then fit as an explanatory variable together with season and interactions. Site within deployment date were included as random effects.

To determine how the relationship between velocity and mobilisation varied across the 3-day period in each season, two mixed-effects models with binomial distributions were fit using the package 'glmmTMB', with rubble transport >1 cm as the response variable (0 or 1), and near-bed wave orbital velocity and day, and their interactions as explanatory variables. Each rubble pieces' unique ID, within site within deployment date, were included as nested random effects. A third and fourth model were fit with identical explanatory variables and random effects, but with the probability of flipping as the response variable for each season. A fifth and sixth model were fit using the package 'nlme' (Pinheiro et al., 2019), utilising a gamma distribution and the same explanatory variables but with 'distance transported by rubble' as the response variable for each season. The response variable was logged to achieve normality. Only rows for which rubble was transported ≥ 1 cm were retained (i.e., zeroes removed) and due to this reduction in replication, the only random effect retained for these models was site.

To determine mobilisation thresholds in the field and investigate the effects of rubble and substrate characteristics on mobilisation, only data from day 1 were used. This is because the day 1 conditions in the field were most like flume conditions, as rubble had been newly deployed and had no opportunity yet to settle. Furthermore, mobilisation in the field was modelled against the full range of velocities pooled across habitats and seasons. A model was fit using the package 'glmmTMB' with the probability of transport > 1 cm as the response variable and velocity, rubble size, branchiness, substrate and all interactions as explanatory variables. Site was included as a random effect. A second model was fit with identical explanatory variables and random effect, but with the probability of flipping as the response variable. To provide a valid comparison to the transport thresholds in the flume, reduced models with velocity as the sole explanatory variable were fit to determine the 50% and 90% thresholds for transport > 1 cm and flipping, averaged across all rubble sizes and substrates. To investigate the distance transported by rubble on day 1, a third model was fit using the "nlme" package with distance as the response variable and velocity, rubble size, branchiness, and substrate as explanatory variables. No interactions were fit due to low replication of rubble pieces that had moved distances > 1 cm. Site was included as a random effect.

Slope was included in each of the three models above but was found to be consistent across rubble size, branchiness and substrate, i.e., there were no interactions with slope when included in full models. Thus, three additional models were fit with only velocity, slope and the velocity:slope interaction as explanatory variables, with movement type as the response variable, and site as the random effect.

## 3    Results

### 3.1    Mobilisation in flume

#### 3.1.1    Loose rubble – Mobilisation thresholds

When averaged across rubble of sizes 4–23 cm, morphologies and substrates, we found that half of all rubble experience rocking motions when velocities reached 0.28 m/s (SE: 0.005), and 90% of rubble rocked at ≥ 0.48 m/s (SE: 0.013). At these higher velocities, pieces were less likely to rock and more likely to be transported or flipped. The 50% and 90% mobilisation thresholds for rubble transport (walk/sliding/flipping) were slightly higher: 0.3 m/s (SE: 0.003); and 0.43 m/s (SE: 0.006), respectively (Table S3). Near-bed wave orbital velocities had to reach 0.34 m/s (SE: 0.004) for 50% of rubble to flip completely, and 0.5 m/s (SE: 0.009) for 90% of rubble to flip (Table S4).

As well as calculating the inertia component for each wave height and period combination in the flume based on the average coral diameter (see 2.1 Methods), we also made these calculations for individual runs using the unique diameter of each piece. Of the cases identified as having the potential for inertia forces to be significant, the majority were runs where rubble did not move. Further, 9.3% (195 of 2,081) were runs where only rocking movements were recorded. The highest velocity represented in these cases was 0.2 m/s, though the majority were much lower (Figure S6). Thus, at velocities <0.2 m/s, there is the potential for inertia forces to contribute to causing rocking motions. But, at a velocity of 0.2 m/s the contribution of inertia is still only 25% of the drag force (not dominant), and the threshold of rocking conditions in the flume, reported above, are drag dominated.

Transport or flipping occurred in only 0.9% of runs where we determined inertia forces to be potentially significant (18 of 2,081 runs) (Figure S7). For these cases, the average contribution of inertia forces to the total force was 36% of the drag force and the highest velocity represented in these cases was 0.16 m/s (Table S7). This indicates that at low velocities <0.16 m/s, there is the potential for inertia forces to be significant. However, this cut-off is well below the 50% and 90% thresholds of transport reported above, and at those velocities the inertia component contributes as little as 0.1% and at most 4.9% to the total force. The threshold of transport conditions in the flume are thus drag dominated.

#### 3.1.2    Loose rubble – Rubble and substrate effects on mobilisation

**Probability of 'rocking'**

Rubble was more likely to rock as velocity increased, but the relationship varied with rubble size, shape, and underlying substrate (Figure 3). Consequently, there were 3-way interactions among velocity, size and branchiness ($\chi^2$ = 55.3, $P$ < 0.001), and among velocity, size and substrate ($\chi^2$ = 17.8, $P$ < 0.001) (Table S5). The branchiness of rubble was an important predictor of rocking. Across all velocities, rubble of all size classes was more likely to rock if they were unbranched rather than branched (except for intermediate rubble 16-23 cm, Figure 3 a, i-iv) (Table S6). Once a velocity threshold was exceeded, rubble size and substrate also played a part. For velocities ≥ 0.2 m/s, the rocking of smaller rubble (4–8 cm and 9–15 cm) was sensitive to the underlying substrate,

being more likely to rock on sand than rubble (Figure 3 a, i-iv) (Table S7). Once velocities exceeded 0.3 m/s, the smallest rubble pieces (4–8 cm) were more likely to rock than all larger-sized rubble (Table S8), averaged across substrate types.

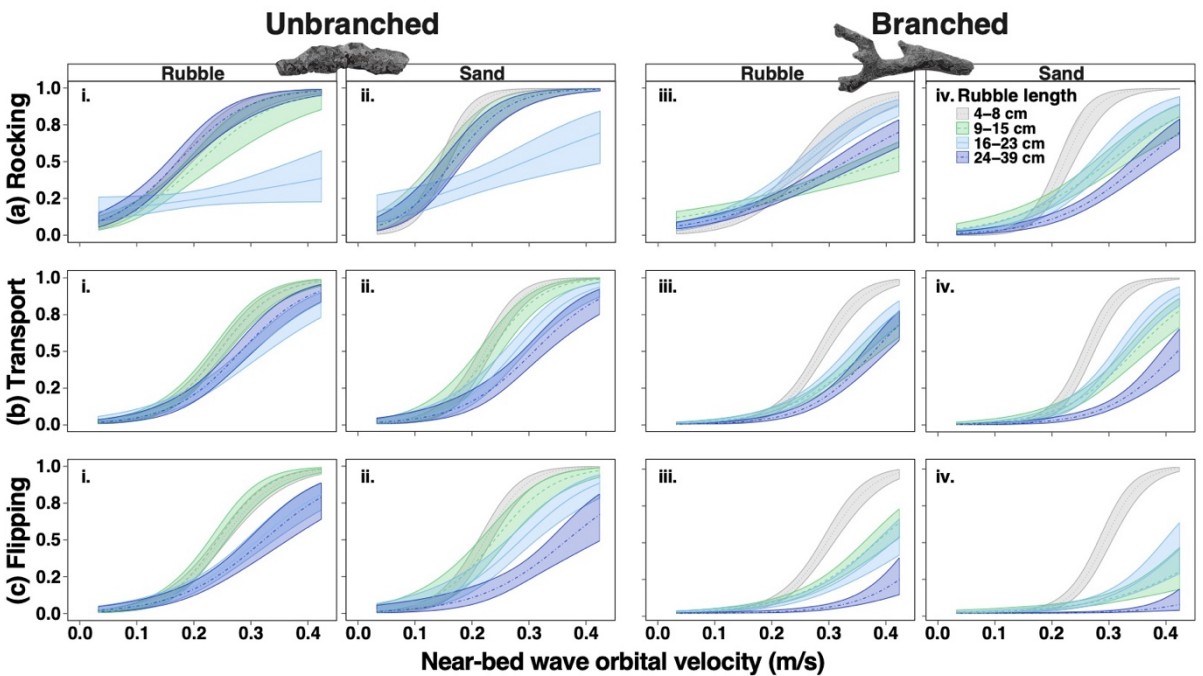

**Figure 3: The probability of (a) rocking, (b) transport, and (c) flipping with increasing near-bed wave orbital velocity for branched and unbranched rubble of four size categories (grey: 4-8 cm; green: 9-15 cm; light blue: 16-22 cm; dark blue 24-39 cm) on rubble and sand substrates. Note that at low velocities <0.2 m/s, we estimate there is the potential for inertia forces to contribute to causing rocking motions; and at velocities <0.16 m/s, there is the potential for inertia forces to contribute to causing transport and flipping.**

**Probability of 'transport' (walk/slide/flip)**

As with rocking movements, the probability of transport also increased with velocity, depending on rubble characteristics and substrate, again with two 3-way interactions (velocity, size and branchiness $\chi^2 = 17.6$, $P < 0.001$; velocity, size and substrate $\chi^2 = 8.9$, $P < 0.03$) (Table S9). Qualitatively, the patterns for transport were similar to those for rocking, but the effect of branchiness changed at high velocities. For example, unbranched rubble was transported more commonly than branched rubble at velocities ≤ 0.4 m/s, after which rubble of both morphologies were equally as likely to be transported, at least for sizes 4–8 cm and 16–23 cm (Figure 3 b, i-iv) (Table S10). Size was a clear predictor of transport, with 4–8 cm rubble more likely to be transported than two groups of larger rubble: 16–23 cm and 24–39 cm, at velocities ≥ 0.2 m/s (Table S11). There was even greater delineation of size if rubble was branched; 4–8 cm branched rubble was more likely to be transported than *all* larger rubble at velocities ≥ 0.3 m/s, on both substrates (Figure 3 b, iii-iv, Table S11). Just as 4–8 cm rubble rocked more easily on sand, it also tended to be transported more easily on sand at velocities ≥ 0.3 m/s. Interestingly, the largest rubble pieces 24–39 cm were more likely to be transported on rubble than on sand at these velocities (Table S12), perhaps due to an ability to sink into sand but not rubble.

**Probability of 'flipping' only**

We distinguish flipping on its own, because it is the form of transport expected to involve some form of abrasion across most surfaces of the rubble. Like rocking and transport probabilities, two 3-way interactions affected the probability of flipping (velocity, size and branchiness $\chi^2 = 18.4$, $P < 0.001$; and velocity, size and substrate $\chi^2 = 10.7$, $P = 0.013$ (Table S13). Again, unbranched rubble was more likely to flip than branched rubble (Figure 3 c, i-iv; Table S14). Yet, branched, small 4–8 cm rubble was much more likely to flip than all larger rubble, particularly at velocities $\geq 0.4$ m/s. Once again branchiness had a strong influence on this relationship, with unbranched rubble pieces having instead *similar* probabilities of flipping across a size range of 4 to 15 cm (Figure 3 c, i-ii) (Table S15). Substrate type had little effect on rubble flipping. However, when pieces started to flip at 0.2 m/s, branched rubble flipped more on rubble substrate than on sand, while unbranched rubble was just as likely to flip on rubble or sand (Table S16).

### 3.1.3 Interlocked rubble

Rubble mobilisation trials were profoundly different when the experimental rubble was interlocked with the second rubble substrate. For interlocked rubble, there was no relationship between velocity and the probability of any type of movement (Table S17). Rubble was very unlikely to move (<7%) even at the highest velocity tested (0.4 m/s). Yet while the probability of any movement was low, when interlocked rubble of both sizes *did* move they most commonly rocked (5 ± 1%) as opposed to being transported (1 ± 0.3%) or flipped (1 ± 0.3%) (rock vs transport: z-ratio = 3.671, $P < 0.001$; rock vs flip: z-ratio = -3.671, $P < 0.001$) (Table S49, Figure S9). In fact, interlocked 4–8 cm rubble was not observed to walk, slide or flip at all.

## 3.2 Mobilisation in field

### 3.2.1 In-situ environment

During deployment periods, higher significant wave heights were recorded in the western monsoon compared to the north-eastern monsoon (Table 2).

**Table 2 Wave statistics for each habitat (aspect and depth) and monsoon season. Mean statistics show average of all 30-minute runs in the 3-day period across 3 sites on the reef flat and 6 sites on sheltered and exposed reef slope (15 sites total). Max statistics show highest of the 30-minute runs. $H_s$ = significant wave height; $T_p$ = peak wave period.**

| Monsoon season | Depth | Aspect | mean $H_s$ (m) | max $H_s$ (m) | mean $T_p$ (s) | max $T_p$ (s) |
|---|---|---|---|---|---|---|
| North-east | 2-3 m | Reef flat | 0.08 | 0.21 | 9.88 | 19.78 |
| | | Southeast (slope) | 0.09 | 0.24 | 9.13 | 14.63 |
| | | West (slope) | 0.11 | 0.27 | 4.52 | 17.31 |
| | 6-7 m | Southeast (slope) | 0.08 | 0.20 | 8.99 | 14.40 |
| | | West (slope) | 0.08 | 0.17 | 3.94 | 8.65 |
| West | 2-3 m | Reef flat | 0.15 | 0.23 | 8.86 | 10.91 |
| | | Southeast (slope) | 0.18 | 0.36 | 10.86 | 19.78 |
| | | West (slope) | 0.18 | 0.74 | 8.90 | 10.98 |

| Monsoon season | Depth | Aspect | mean $H_s$ (m) | max $H_s$ (m) | mean $T_p$ (s) | max $T_p$ (s) |
|---|---|---|---|---|---|---|
| | 6-7 m | Southeast (slope) | 0.17 | 0.33 | 10.65 | 19.57 |
| | | West (slope) | 0.16 | 0.72 | 8.89 | 11.61 |

Corresponding near-bed wave orbital velocities also were significantly higher in the western monsoon than the north-eastern monsoon, except for reef flat and exposed shallow slope sites (despite a trend, Figure 4, Table S18, S20).

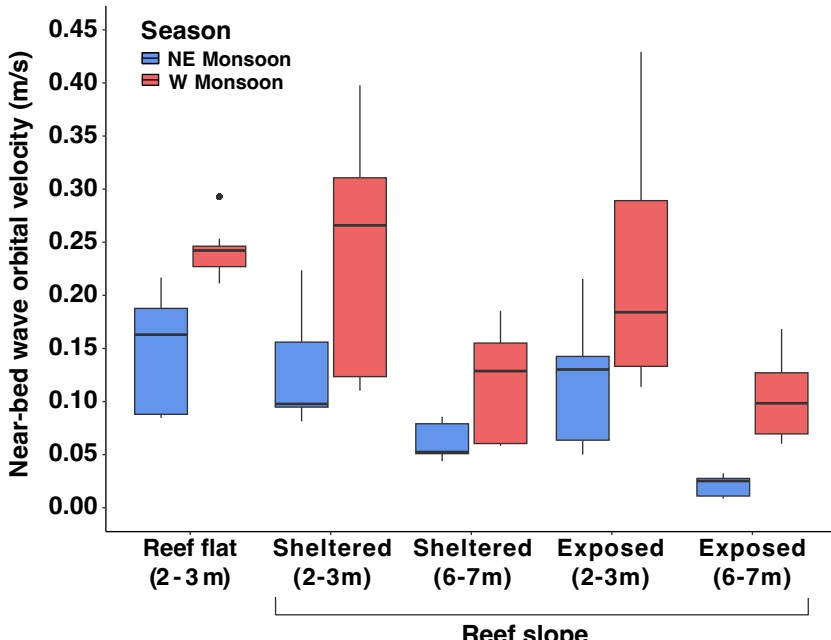

**Figure 4: Boxplots for the 9 (1 per day x 3 days x 3 sites) fastest near-bed wave orbital velocity values estimated for each habitat in each monsoonal observation period.**

Consequently, there was an interaction between season and habitat on near-bed wave orbital velocity ($\chi^2 = 102.2$, P < 0.001, Table S19). In both seasons, shallow reef slope sites (2-3 m) experienced faster velocities on average than deeper sites (6-7 m) (Table S21). Curiously, the velocity did not vary significantly between sheltered and exposed sites. However, the exposed shallow reef did experience the greatest wave height and fastest velocity in both seasons (Figure 4, Table 2).

### 3.2.2    Mobilisation across 3-day deployments

The relationship between velocity and rubble mobilisation across days was investigated for each season separately.

In the western monsoon, rubble was more likely to be transported and more likely to be flipped as the velocity increased, but only on day 1, resulting in an interaction between day and velocity (transport: Figure 5 a, $\chi^2 = 11.3$, $P = 0.004$; flipping: Figure 5 b, $\chi^2 = 7.416$, $P = 0.025$) (Table S22/S23). For example, the probability of transport

increased from 30% to 60% moving from 0.1 to 0.4 m/s on day 1, but on day 2 these velocities both yielded only a 20% chance of transport (Table S24). As for the likelihood of transport and flipping, rubble travelled slightly greater distances as velocity increased ($\chi^2 = 7.1$, $P = 0.008$), and travelled on average 1.6 cm more on day 1 than day 2 during the western monsoon (Figure 5 c, Table S26/S27).

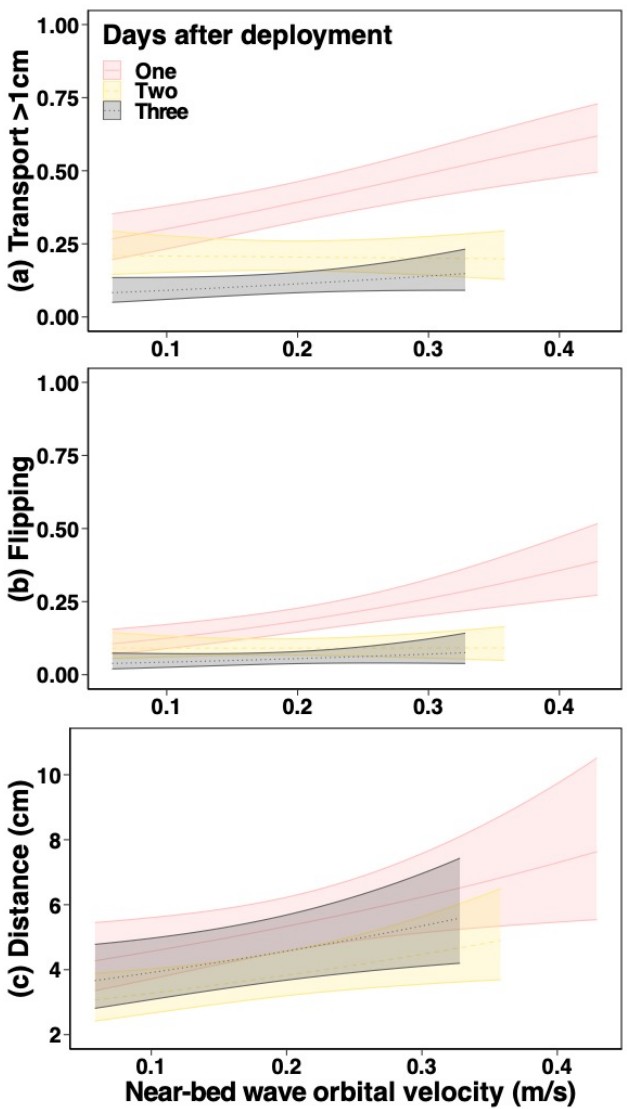

**Figure 5: Relationship between near-bed wave orbital velocity (m/s) and (a) the probability of rubble transport (> 1cm), (b) probability of flipping, and (c) distance transported, on each day of the 3-day periods during the western monsoon (averaged across habitat).**

In the north-eastern monsoon, there was no relationship between velocity and rubble transport nor flipping, because the range of velocities captured in this season was comparitively narrower (Figure 4). However, there was an effect of day on the probability of transport ($\chi^2 = 7.304$, $P = 0.026$, Table S28) and flipping in this season ($\chi^2 = 28.1$, $P < 0.001$, Table S29). At the mean velocity in the north-eastern monsoon (0.1 m/s), the probability of flipping on day 1 was 13%, and fell on days 2 and 3 to only 6% (Table S31). Rubble also travelled shorter distances on day 3 than day 1 (z-ratio = 3.9, $P < 0.001$, Table 32/33).

### 3.2.3 Mobilisation thresholds

The mobilisation thresholds in the field were estimated using rubble movement data for day 1 only (as the most representative scenario to the flume trials, i.e., rubble pieces were newly deployed and not 'settled') and using data from both seasons (to capture a wider range of velocities). The 50% and 90% mobilisation thresholds for transport (> 1 cm) in the field, averaged across all rubble sizes (4–23 cm), branchiness and substrate characteristics, were 0.30 m/s (SE: 0.037) and 0.75 m/s (SE: 0.146), respectively, on day 1 (Table S34). We note however that the 90% threshold for transport is above the range of velocities measured in the field and should thus be considered cautiously compared to the 50% threshold. We do not report the 50% or 90% thresholds for flipping in the field for the same reason.

### 3.2.4 Rubble and substrate effects on mobilsation

**Probability of 'transport' (walk/slide/flip)**

To investigate the effects of rubble and substrate characteristics on the relationship between velocity and mobilisation in the field, data were also used from both seasons on day 1.

The probability of rubble transport (> 1 cm) on day 1 increased with velocity, but this relationship varied among rubble sizes ($\chi^2$ = 10.039, $P$ = 0.007) (Figure 6 a, Table S35). At lower velocities, small, 4–8 cm rubble was transported more commonly than medium rubble, 9–15 cm, which moved more than large rubble, 16–23 cm. In the field, rubble of all sizes was equally likely to be transported at velocities $\geq$ 0.3 m/s (Figure 6 a; Table S36), in contrast to the flume trials where smaller rubble always moved more than larger pieces across increasing velocities. Like the flume trials, rubble branchiness had a clear effect on rubble transport in the field, with unbranched rubble 1.7 times as likely to be transported as branched rubble (when averaged across velocity, substrate and size) (Table S37). The substrate type did not influence rubble transport in the field study ($\chi^2$ = 0.4, $P$ = 0.80) (Table S35).

The relationship between velocity and transport changed with the steepness of the slope ($\chi^2$ = 5.6, $P$ <0.001) (Table S38). For flatter areas, rubble was more likely to be transported as velocity increased, whereas on steep slopes, the probability of transport did not increase by as much (Figure 6 c).

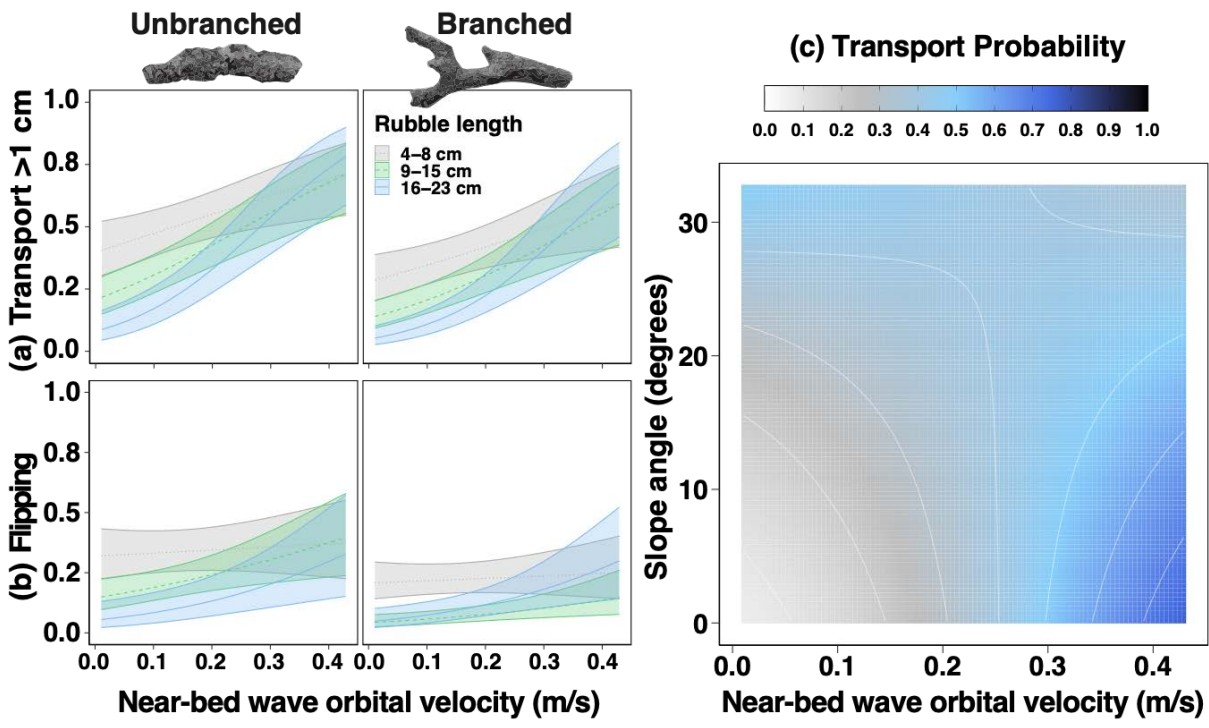


**Figure 6: Relationship between near-bed wave orbital velocity (m/s) and the (a) probability of rubble transport (> 1 cm), (b) probability of flipping for each rubble size and branchiness type, and c) how the slope angle and near-bed wave orbital velocity affects the probability of movement of rubble pieces.**


For example, at velocities of 0.1 m/s and on very gentle slope angles of 3° (common on the reef flat), just 16% (± 2.7%) of rubble would be transported, compared to 33% (± 2.6%) of rubble on 22° (steep) slopes, common at
deep reef slope sites (Table S39). When water velocity increased to 0.4 m/s, rubble had a 69% (± 7.8%) chance and 48% (± 1.1%) chance of moving on very gentle and steep slopes, respectively (Figure 6 c). At velocities
≥ 0.2 m/s, there was no significant difference in the probability of transport across slope angles (Table S39).

**Probability of flipping only**

In the field, rubble was less likely to be flipped entirely than to be transported (Figure 6 b). As with the pattern observed for rubble transport, unbranched rubble flipped more commonly than branched rubble. However, unlike
rubble transport, unbranched rubble only flipped more than branched rubble when they were small to medium, i.e., 4-15 cm in length ($\chi^2 = 8.3$, $P = 0.015$) (Table S40/41). Larger rubble of length 16–23 cm had a relatively low
probability of flipping regardless of branchiness. Small (4–8 cm) rubble flipped more often than rubble sized from 9 to 23 cm (Table S42). However, as for transport, flipping became less dependent on rubble length as velocity
increased, and all sizes were equally likely to flip at velocities ≥ 0.4 m/s ($\chi^2 = 7.2$, $P = 0.03$) (Table S40/S43).

Also, similarly to transport, the probability of flipping did not appear to vary with the substrate type ($\chi^2 = 4.9$, $P$

$= 0.083$) (Table S40). Furthermore, while slope angle had some effect on the probability of transport, it did not appear to affect the probability of rubble flipping in the field ($\chi^2 = 0.4$, $P = 0.536$) (Table S44).

**Distance transported**

The distance travelled by rubble increased with velocity ($\chi^2$ =12.3, $P$ <0.001) but was not affected by rubble size or branchiness (Table S45). Substrate type, however, did affect the transport distance ($\chi^2$ =6.2, $P$ = 0.046). Just as
smaller rubble moved more easily on sand in the wave flume, rubble travelled slightly further on sand (6.2 ± 0.8 cm averaged across velocities) than on rubble (4.7 ± 0.4 cm) over the course of one day (t-ratio = -2.3, $P$ =
0.05, Table S46).

As for transport probability, there was an interaction between velocity and slope for distance travelled ($\chi^2$ = 26.2,

$P$ < 0.001) (Table S47). At low velocities, rubble travelled greater distances as the steepness of the slope increased, likely aided by gravity. For example, on very gentle slopes (3°) rubble moved less distance (3 ± 0.2 cm) than
rubble on very strong (22°) slopes (5 ± 0.3 cm) at velocities of 0.1 m/s. Rubble travelled further as velocity increased on very gentle slopes (e.g., 22.9 ± 9.2 cm on 3° slopes at 0.4 m/s), but this pattern wasn't observed on
steeper slopes at the same velocity (e.g., 3 ± 0.5 cm on 22° slopes) (Table S48).

**4    Discussion**

Here we characterised the physical parameters (i.e., near-bed wave orbital velocity, substrate type, reef slope angle) that influence rubble mobility in a flume and field setting across a range of rubble sizes and
morphologies. As near-bed wave orbital velocity increased, rubble was more likely to rock, be transported and travel greater distances. Across flume and field environments, small and/or unbranched rubble pieces were
generally mobilised at lower velocities than larger, branched rubble, while reef slope angle and substrate (sand or rubble) had more nuanced effects. Averaged across rubble and substrate types, the 50% rocking threshold
was slightly lower than the 50% transport thresholds, which were almost identical between flume and day 1 field results. Interlocking and 'settling' of rubble was a strong inhibitor of mobilisation, especially of transport.
Interlocked rubble in the flume had only a 7% chance of moving, and in the field, the likelihood of rubble mobilisation decreased over the course of the 3-day deployments. We hypothesise that rubble experienced
'settling' or short-term stabilisation, whereby pieces were less likely to be transported on days 2 or 3 than day 1 at the same velocity. While the field results show rubble is capable of being mobilised during average wave
conditions across the normal tidal cycle, if the rubble settling effect is significant in an area, specific storm events that cause higher velocities are likely to be more influential to mobilisation.
In the wave flume and in the field, 50% of loose, cylindrical rubble ranging from 4–23 cm was transported at 0.3 m/s. Similar velocities to the reported thresholds have been observed on coral reefs globally, suggesting that
rubble could be shifted under ambient conditions, depending on substrate, rubble typology and interlocking. Near-bed wave orbital velocities > 0.3 m/s have been reported on coral reefs in the Great Barrier Reef (Harris et
al., 2015), Palmyra Atoll (Rogers et al., 2015; Monismith et al., 2015), Moorea (Monismith et al., 2013) and Puerto Rico (Viehman et al., 2018) and are likely common in nearshore and surf zone settings on reef-slope,
crests and flats. Wave and tide-induced current velocities above 0.3 m/s are likely found on most coral reefs, but not all reef environments (Kench, 1998a; Helmuth and Sebens, 1993; Sebens and Johnson, 1991). Threshold
wave-orbital velocities in the present study are comparable to the modelled initiation of motion thresholds for rubble treated as simplified rectangular prisms with dimensions drawn from mean-sized rubble (length: ~3.3 cm,

up to 10 cm) at a ship-grounding site on the south coast of Puerto Rico (Viehman et al., 2018). Reported wave-orbital thresholds were ~0.09-0.2 m/s for sliding and ~0.12-0.34 m/s for flipping, depending on rubble size and the degree of flow blocking by grouping. The thresholds reported in the present study differ in that they consider a wider range of rubble lengths and shapes, are observational as opposed to modelling based, and are described in terms of probability rather than absolute initiation of motion.

The frequency at which rubble is transported (the transport return interval) will affect the length of stable periods or windows of recovery for coral recruitment and binding. Using hindcast wave modelling, Viehman et al. (2018) revealed the return interval for rubble sliding and overturning at their site in Puerto Rico was 7 and 12 days, respectively, with some, but not all, hindcast events aligning with tropical storms and cyclones (Viehman et al., 2018). Similarly, Cheroske et al. (2000) showed that rubble pieces tumbled on average about once every 15 days in Kaneohe Bay, Hawaii. However, the maximum flow speeds in the Kaneohe Bay study were relatively high, 0.6-1.5 m/s (Morgan and Kench, 2012), compared to flows up to 0.43 m/s at Vabbinfaru Reef. Owing to the protection afforded from storms and swell due to its location inside North Male' Atoll (Rasheed et al., 2020), we expect longer average return intervals on Vabbinfaru Reef. For example, islands <5 km (Dhakandhoo) and 15 km (Hulhudhoo) from the western edge of nearby South Maalhosmadulu Atoll experience 60% and 80% reductions in wave height, respectively, compared to mean incident ocean swell (Kench et al., 2006; Young, 1999). Higher energy movement events in the Maldives are likely driven more commonly by monsoonal wind patterns, and clustered in the western monsoon. For example, during the north-eastern monsoon, a velocity of 0.3 m/s (expected to transport 50% of rubble pieces in the field) was never exceeded in 37 observed days, and in the western monsoon, it was exceeded on 4 of 32 days, at shallow sites only, with velocities exceeding 0.4 m/s on only 1 day at an exposed shallow site in the western monsoon. Considering wind speeds and direction during observational periods for each monsoon are typical of respective conditions over the past 33 years (Figure S3), this indicates a transport return interval of ~8 days, but only at shallow sites during the western monsoon. Furthermore, we maintain that the return interval is likely to be much longer than this, considering that transport thresholds increase as rubble 'settles' over time and as organisms such as sponges, bryozoans and CCA bind rubble (Kenyon et al., 2023). However, while a window where conditions are too calm for transport is good for coral recruitment, binding between rubble pieces could yet be prevented by rocking motions, particularly for small, unbranched pieces (e.g., 50% of loose, 4-8 cm unbranched rubble predicted to rock at 0.18 m/s in the flume across substrates; Figure 3a). Thus, we conservatively estimate that recovery windows for binding are likely to occur during the calmer north-eastern monsoon, when wave energy impacting the atoll is less and wave heights are smaller (Kench et al., 2006).

Curiously, at the same wave orbital velocity, the probability of rubble transport was lower in the north-eastern monsoon than in the western monsoon, suggesting there is greater complexity driving rubble transport than has been captured. For example, while the velocity across the day might be similar, sites in the western monsoon may have experienced a higher frequency of similar velocities throughout the day, providing more opportunities for mobilisation (supported by sites in the western monsoon having higher daily-average wave orbital velocities, as well as higher maximum velocities – Figure S4). Alternatively, the greater hydrodynamic energy in the western monsoon may have primed the substrate to better facilitate transport. Even within the western monsoon, however, the probability of mobilisation decreased by ~10% each day over the three days (velocity dependant). Rubble may have 'settled' into more stable positions after being moved from the position in which they were placed by divers on day 1. Several rubble pieces shifted into crevices, particularly in shallow reef slope sites where hard carbonate

and coral created a more structurally complex substrate than sandier, deeper slopes (T Kenyon, *pers. obs.*). On One Tree Island, Thornborough (2012) found branching rubble was regularly lodged under plate or boulder rubble
or interlocked together into a rubble ridge within six days of the commencement of experiments. There, interlocked plate rubble also remains stable under energetic, tidally-driven conditions (Thornborough, 2012).
Presumably, higher velocities would be required to move rubble that has a) settled deeper into the substrate by downward flow forcing, or b) wedged against a surface by lateral flow forcing. In the present study, some rubble
still moved after settling on days 2 and 3, but manually interlocked rubble in the flume was very unlikely to be transported even at the maximum velocity of 0.4 m/s. Higher energy, variable wave environments would likely
foster more unstable rubble beds than lower energy, constant wave environments, where rubble has time to settle. In these more energetic and/or variable settings, and with smaller, simpler-shaped pieces, rubble may not settle
and/or interlock routinely, and could persist as an unstable bed for decades (Fox et al. 2019).

As expected, the threshold for rubble mobilisation varied according to rubble branchiness, in both controlled

and reef environments. Generally, unbranched rubble was more likely to rock, walk, slide or flip, than branched rubble. Branches can stabilise the rubble piece by digging into the sand or wedging against or beneath another
rubble piece, thus explaining why living coral fragments with branching morphologies have increased post-breakage survival compared to those with non-branching morphologies (Tunnicliffe 1981; Heyward and Collins
1985; Smith and Hughes 1999). Branched fragments and rubble would become lodged more easily in crevices or interlock together to form stable rubble beds, which can act as platforms for coral recruitment (Aronson &
Precht 1997). Size also affected the likelihood of mobilisation of rubble, reflecting studies on live fragment mobilisation and survival (Smith and Hughes, 1999; Hughes, 1999). Regardless of whether they had branches or
not, small cylindrical rubble (particularly 4–8 cm) were more likely to be transported than larger pieces. However, size only influenced rubble transport in the field up to velocities of 0.3 m/s. Regardless, interventions
might be considered at lower transport thresholds (e.g., 50% of loose, 4-8 cm unbranched rubble predicted to move at 0.14 m/s in the field; Figure 6a) if a rubble bed is comprised predominantly of very small pieces, which
is more commonly the case with anthropogenic disturbances such as ship groundings, human trampling and blast fishing (Kenyon et al., 2023). In Japan for example, rubble mounds formed seaward of coastal armouring
were lower in weight, length, and surface complexity than rubble from natural beds (Masucci et al., 2021).

We expected rubble to move more easily over sand, as shown previously (Heyward and Collins 1985; Bruno

1998; Bowden-Kerby 2001; Prosper 2005). However, substrate type had little effect on rubble mobilisation in the flume, except that small rubble were more likely to rock and be transported on sand than on rubble once velocities
exceeded 0.2 m/s. In the field, although the distance travelled by rubble was slightly higher on sand than on rubble substrates, no effect of substrate on mobilisation probability was observed. This is potentially owing to the limited
available sandy areas free of rubble on which to conduct trials as a consequence of the severe coral bleaching in the Maldives in 2016 (Perry and Morgan, 2017), leading to a mixed rubble-sand substrate. Greater distinction
between substrates may have been observed in the flume if the first rubble substrate was comprised of larger-sized pieces more capable of 'snagging' and interlocking the experimental pieces. The trials with the second
rubble substrate demonstrated how interlocking provides a significant impediment to mobilisation. After an intense disturbance on a healthy reef, there is likely to be more rubble (multiple layers) and a greater proportion
of rubble resting on other rubble, which – depending on branchiness and rubble size – could facilitate interlocking (Aronson and Precht 1997). For smaller quantities of rubble, the rubble bed might be thinner (perhaps only one
layer), and more rubble will be in contact with sand or hard carbonate substrate underneath, with less capacity for interlocking.
There were instances of rubble transport in the field even when the highest estimated velocity was ~0.01 m/s. Several video observations of deployed rubble indicated no disturbance by fish and invertebrates, but this cannot
be ruled out completely (Ormond and Edwards, 1987). Rubble movement on steeper sections of the slope were aided by gravity. In fact, all instances of movement at velocities <0.05 m/s occurred in steep 6-7 m slope sites.
Hughes (1999) found that fragments moved downslope in the absence of any major storms, most likely due to gravity-driven hillslope processes observed in marine and terrestrial systems (Salles et al., 2018). At lower
velocities (< 0.1 m/s) rubble was aided by gravity and more likely to move and travel further on steeper slopes than flat and gentle slopes. Yet, as water velocity increased, rubble travelled shorter distances on steeper slopes.
It is possible that higher velocities are indicative of waves with greater asymmetry that oppose gravitational transport and therefore maintain rubble at higher positions on the slopes, similar to the concept of equilibrium
position of sediment on beach shorefaces over time (Ortiz and Ashton, 2016). While no significant relationship was detected between wave orbital velocity and direction, there was a trend in this direction. At shallow reef slope
sites, which experienced higher velocities, ~19% of rubble movements were upslope, compared to just ~3% at deeper sites. Given the size of rubble, substantial upslope movement likely requires storm energy (Woodley et al.,
1981b; Harmelin-Vivien and Laboute, 1986). Rubble might also travel further on flatter slopes at high velocities as a result of the association between slope and depth, i.e., flat and gentle slopes found primarily in reef flat and
shallow sites; steep slopes primarily in deep sites. Reef flat and shallow slope sites experienced higher average velocities than deeper sites (Figure S4), and thus experienced a higher frequency of velocities close to the
maximum, providing more opportunities for mobilisation. Understanding the links between hydrodynamics and bathymetry of a disturbed reef is evidently important in determining its vulnerability to rubble mobilisation and
recovery potential.

  Two important factors to be considered in context of the present study are the density or crowding of the rubble,

and the effect of rubble age on mobilisation thresholds. Following a disturbance, rubble will become increasingly distinct from recently-killed coral in size, porosity, density and surficial encrustation, which will affect its
hydrodynamic behaviour (Allen, 1990). Rubble is prone to further mechanical breakdown over time, due to incidental bioerosion by predators and grazers, and direct bioderosion by borers (Scoffin 1992, Perry and Hepbum
2008), which may be exacerbated under certain environmental conditions, e.g., high nutrients and/or depth (Hallock, 1988; Pandolfi and Greenstein, 1997). Initially, rubble is expected to become less dense and more
porous, as bioeroders and borers infiltrate the dead skeleton, although the time-frames for these processes are largely unknown (but see Pari et al. 2002; Tribollet et al. 2002). The skeletal density of rubble used in the wave
flume was $2.2 \pm 0.1$ g/cm$^3$ (mean $\pm$ SE) and on the reef was $1.9 \pm 0.04$ g/cm$^3$ (mean $\pm$ SE), which is similar to the mean coral skeletal density reported from a previous study at Vabbinfaru ($1.85$ g cm$^{-3}$) (Morgan and Kench,
2014b), suggesting that it had not been heavily bioeroded. Over time and with encrustation by coralline algae and in-filling of sediments into pores, cementation by magnesium calcite and aragonite could increase density (Scoffin
1992), also affecting mobilisation thresholds. The bioerosional potential and subsequent mobilisation thresholds of rubble vary across different rubble morphologies and zones. Bioerosional processes proceed more readily in
deeper, lower energy environments, and in more dense, massive morphologies compared to branching rubble, likely due to their higher residence times in active bioerosion zones (Pandolfi and Greenstein 1997, Greenstein

and Pandolfi 2003, Perry and Hepbum 2008). The density of branching coral rubble might remain higher than massive coral rubble, resulting in higher velocity thresholds (Pandolfi and Greenstein, 1997). Yet, branching morphologies are also more prone to breakage, leading to smaller pieces and subsequently more movement.

Mobilisation thresholds will also be affected by how many rubble pieces are in a rubble bed. Notably, thresholds are likely to be lower for individual pieces, used in the current study, as they are exposed to flow on all sides. Densely packed rubble is likely to be more stable than individual pieces, even without interlocking, due to the protection afforded by surrounding rubble. Similar considerations are made when assessing transport of boulders surrounded by rock on the lee side of flow, which have a higher threshold of motion than free (not surrounded) boulders (Nott 2003, Nandasena et al. 2011). In modelling the mobilisation thresholds of oblong-shaped rubble exposed to flow, Viehman (2018) applied a blocking factor to vary the amount of rubble area exposed to flow because of varying degrees of crowding (Storlazzi et al. 2005). Surprisingly, this factor resulted in only very slight variations in the sliding and overturning thresholds. Tajima and Seto (2017) reported that most pieces in coral gravel beds shifted at 0.25-0.5 m/s, a comparable threshold to that reported for rubble pieces here, yet pieces in these gravel beds were small, only up to 2 cm. Mobilisation of beds of larger-sized rubble common on coral reefs should be investigated in further trials in a controlled wave flume environment. Individual pieces in moveable, natural rubble beds could be tagged and tracked over longer periods to further understand mobilisation as a group.

## 4.1 Implications for management

The scale of reef degradation and subsequent intervention methods is vast, putting pressure on reef restoration budgets. While operationalising the implementation of reef restoration at scale is investigated (Saunders et al., 2020), tools that allow managers to prioritise reefs that are particularly vulnerable to rubble mobilisation, and thus longer natural recovery times, are essential (Kenyon et al., 2023). The results of this study provide information toward improved management of damaged reefs with high rubble cover. Broadly, rubble stabilisation interventions might be considered at lower mobilisation thresholds if a rubble bed is composed mostly of loose (not interlocked), small pieces, particularly with low morphological complexity, which is more commonly the case with anthropogenic disturbances such as ship groundings, human trampling, coastal armouring and blast fishing (Masucci et al., 2021; Kenyon et al., 2023). Importantly, groove sites can also be characterised by these rubble types (Wolfe et al., 2023), but should not be considered for interventions because they are geomorphological features with hydrodynamic conditions commonly driving rubble entrainment and deposition (Shannon et al., 2013; Duce et al., 2022). More comprehensively, the mobilisation estimates reported here can be used in modelling frameworks that predict the frequency of everyday rubble mobilisation in a certain location, based on a modelled time series of wave climate estimates, such as the everyday wave conditions model developed for the Great Barrier Reef (Roelfsema et al., 2020). Reefs or areas of reefs at higher risk of frequent rubble mobilisation can be prioritised for rubble stabilisation interventions following disturbances, with predictions being improved through consideration of the mobilisation processes discussed, e.g., settling and interlocking over time; bathymetry (e.g., slope, geomorphology); rubble quantity, size and morphology (driven by disturbance, surrounding coral cover and diversity); water quality and bioerosion.

**Acknowledgements**

This study was conducted in collaboration with the Banyan Tree Marine Laboratory. In-kind contributions were received from Banyan Tree Vabbinfaru and Angsana Ihuru, including the Dive Centre headed by Mujuthaba Ali. We wish to acknowledge Mohamed Arzan, Zim Athif, Amal Charles Everitt, Samantha Gallimore, Danielle Robinson, Crystle Wee, Toby Mitchell, Ahmed Tholal, Ali Nasheed, Jason Van Der Gevel, Stewart Matthews, Ananth Wuppukondur, Matthew Florence and Nick Brilli for assistance in the field and laboratory. This study was funded in part by a PADI Foundation Grant and GBRMPA Science for Management Award to T. M. Kenyon, and ARC grants to P. J. Mumby. Support was also provided by an Australian Government Research Training Program (RTP) Scholarship (stipend), and from the Australian Government's Reef Restoration and Adaptation Program. Limited Impact Accreditation No. UQ005/2016 used for the collection of rubble used in the flume.

**Author contribution**

TMK, DH, CD, PJM conceived field experiments; TMK, TB, DC, PJM conceived flume experiments; TMK conducted flume and field work, processed and analysed data, wrote text; DH, TB, DC, CD, GW, SN, PJM contributed and edited text; DH, CD, GW, SN, PJM provided supervision.

**Competing interests**

The authors have no competing interests to declare.

**Code/Data availability**

Datasets and code available at https://github.com/TMKenyon/rubblemobthresholds.git

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
