# Peer review of "Mobilisation thresholds for coral rubble and consequences for windows of reef recovery"

_Biogeosciences, 2023_

## Author Response (AR1)

**RC1**: 'Comment on bg-2023-2', Anonymous Referee #1, 14 Mar 2023  reply

General comments:

This well-written study presents work empirically determining the threshold of mobilisation
for individual rubble pieces of varying shapes and sizes, and on different substrate types
and slopes, in both controlled and field settings (the Maldives). Rubble movement is relevant
because it impacts coral recovery, and many threats to coral (e.g. destructive fishing,
storms, bleaching) result in coral breakage and/or death. These pieces become "rubble" of
various sizes and shapes.  The experiments are clever and thorough, designed to elucidate
the probability of rubble 'rocking' or various types of 'transport' (walk/slide/flip).
Unsurprisingly, the authors found similar mobilisation thresholds in the wave flume and in
the field, and that the probability of rubble mobilisation increases with higher velocity. Also
as common sense and previous work would suggest, it decreases as: (i) rubble size
increases; (ii) morphological complexity/'branchiness' (of both the rubble and of the
substrate type) increases; and (iii) as the slope angle decreases (and the contribution of
gravity subsequently decreases). Interlocking and 'settling' of rubble was a strong inhibitor
of mobilisation.

Specific comments

While the authors did find some some nuanced results (e.g. larger rubble is more likely to
settle into sand) and differences between the northeastern and western monsoon seasons,
overall, their results seem a very sophisticated experimental demonstration of what common
sense would predict. While sentences in the Abstract (l 19-21) and Introduction (l 37-40)
suggest relevance for managers of reefs that exhibit a significant increase in rubble cover,
there is no discussion of what managers can actually do once they have the information
presented herein. While there is mention of "rubble stabilisation interventions to enhance
coral recruitment and binding," it's unclear that the results from this work would actually be
needed to predict the likelihood of natural rubble stabilisation and recovery **beyond simple**
**first principles**. I suggest the discussion at least address potential management relevance,
including context for discussions for rubble stabilization, budget needed vs. scale of the
problem, etc.

No technical corrections.

Our response:

We thank Reviewer 1 for their review of the manuscript and are pleased that they found the
results to be novel yet intuitive. We thank the reviewer for their comment regarding
highlighting potential management relevance, which is a valuable addition to the
manuscript. The revised manuscript includes reference to management relevance in the
introduction (as before), and in the discussion, pasted below and on line 642 of the revised
manuscript (no track changes).

**"Implications for management**

"The scale of reef degradation and subsequent intervention methods is vast, putting pressure on reef restoration
budgets. While operationalising the implementation of reef restoration at scale is investigated (Saunders et al.
2020), tools that allow managers to prioritise reefs that are particularly vulnerable to rubble mobilisation, and thus
longer natural recovery times, are essential (Kenyon et al. 2022). The results of this study provide information toward improved management of damaged reefs with high rubble cover. Broadly, rubble stabilisation interventions
might be considered at lower mobilisation thresholds if a rubble bed is composed mostly of loose (not interlocked),
small pieces, particularly with low morphological complexity, which is more commonly the case with
anthropogenic disturbances such as ship groundings, human trampling, coastal armouring and blast fishing
(Masucci et al. 2021, Kenyon et al. 2022). More comprehensively, the mobilisation estimates reported here can
be used in modelling frameworks that predict the frequency of everyday rubble mobilisation in a certain location,
based on a modelled time series of wave climate estimates, such as the developed everyday wave conditions model
for the Great Barrier Reef (Roelfsema et al. 2020). Reefs or areas of reefs at higher risk of frequent rubble
mobilisation can be prioritised for rubble stabilisation interventions following disturbances, with predictions being
improved through consideration of the mobilisation processes discussed, e.g., settling and interlocking over time;
bathymetry; rubble quantity, size and morphology (driven by disturbance, surrounding coral cover and diversity);
water quality and bioerosion."

**Citation**: https://doi.org/10.5194/bg−2023−2−RC1

**RC2**: 'Comment on bg−2023−2', Anonymous Referee #2, 31 May 2023  reply

General comments

This manuscript describes an interesting study of movement of coral rubble under waves.
The lab and field measurements of rubble movement seem to have been generally well-
executed, the combination of lab and field measurements is informative, the figures show
interesting patterns, and the datasets have a lot of potential. While I think the manuscript
has potential to ultimately be a nice contribution, I do have serious concerns about aspects
of the analysis and the way some of the methods and results are presented in this
submission. My major concerns are around the treatment of wave period, which is very
different (factor of up to 10) in the lab versus in the field, and the use of orbital velocity as
the hydrodynamic parameter against which rubble movement is plotted and assessed.

The physics of rubble motion under waves isn't fully explained in the manuscript so I
provide some background here. The total force on an object under waves is the sum of two
components: the inertial force and the drag force. The drag force is proportional to orbital
velocity squared and dominates only if the orbital excursion is substantially larger than the
size of the object (Keulegan Carpenter number KC>1). From my back−of−the−envelope
calculations this seems to be the case in much of the field data presented. However, if the
orbital excursion is smaller than the object size (KC<1), the inertial force is the dominant
force on the obstacle. The inertial force is proportion to the fluid ACCELERATION, which is
the orbital velocity multiplied by the wave frequency (2*pi /PERIOD). By my calculations the
inertial force should be the dominant force for many of the lab flume conditions. The wave
period is therefore a critical parameter for this problem, in addition to the orbital velocity.

Because of the very different wave periods between the lab and the field, comparisons
between the two datasets need to be done very carefully/cautiously. Additionally, in the lab
flume, it seems the period was changed (somewhat arbitrarily when breaking was observed),
but the combinations of wave height and period are not reported in the manuscript. A table
of the combinations of conditions in the lab flume experiments needs to be reported, along
with corresponding bottom orbital excursions, velocities, accelerations. This will allow
comparison of orbital excursions with rubble sizes which will inform as to whether drag
(proportional to velocity squared) or inertial force (proportional to acceleration) is the
relevant force. Ideally, the probability of movement would be plotted against a measure of
the total force rather than velocity. There is a nice paper by Viehman et al. (2018) that lays out these forces on rubble. It is cited briefly in the introduction, but I think it could be a
useful reference for sorting out this issue of dominant forces.

The figures are generally well-constructed and the manuscript text is well-organized and
generally well-written.

I provide a few specific comments below, but I have not provided line-by-line comments at
this point because of the critical major issues described above.

Our response:

We thank Reviewer 2 for their review of the manuscript and the time taken to provide
comments. We thank the reviewer for their queries in relation to the methodology, which
has led to an improved manuscript through inclusion of additional information, though we
note that the results do not change substantially. We believe the review comments regarding
inertial forces are due to an omission on our part in the original manuscript, where the coral
diameter was never stated. Consequently, we believe the reviewer may have anticipated
significant inertia forces in the laboratory assuming coral diameters of 0.04–0.2 m, instead
of coral diameters of 0.01–0.02 m. The reviewer is correct that coral diameters of the
former size range would lead to significant inertia forces. We apologise that the coral
diameter was not previously included in the original manuscript, but rather only the lengths
were specified.

The actual coral diameters used in the lab do not lead to significant inertia forces for the
wave conditions that lead to rubble movement. This is addressed at length in these
comments, where we have derived a new relationship for the contribution of the inertia
force to the total maximum force as a proportion of the drag force. We have included a table
of wave conditions used in the flume (Table S1 – line 35) and the field (Table S2 – line 42) in
the revised Supplementary Material. These show the average coral rubble diameter,
significant wave height, period, water depth, and corresponding velocities, inertia force
component and bottom orbital excursions for all wave conditions used in determining the
relationship between velocity and movement.

These tables highlight in which conditions there is potential for the inertia force to be the
dominant force as opposed to drag, based on an average coral diameter of 1.64 cm (range
~1–2 cm) in the flume and 1.69 cm (range ~1–3 cm) in the field. The calculations are
outlined below.

Assuming the drag and inertia coefficients have the same magnitude, the ratio of the
maximum inertia force to the maximum drag force is given by Dean and Dalrymple (1991)
as:

$\frac{F_I}{F_D} = \frac{\pi^2}{KC} = \alpha$     where $KC = \frac{uT}{\varnothing}$ ; KC = Keulegan-Carpenter number , u = maximum orbital
wave velocity, T = wave period $\varnothing$ = rubble diameter

Hence $F_I = \alpha F_D$

The maximum total force is again given by Dean and Dalrymple (1991), noting that the drag
and inertia forces are out of phase,

$F_T = F_D + \frac{F_I{}^2}{4.F_D}$          *$F_T$ = maximum total force, $F_D$ = drag force, $F_I$ =inertia force*

*which can be written as*

$F_T = F_D + \frac{\alpha^2}{4} F_D$

*or*

$F_T = F_D + \frac{24}{KC^2} F_D$

The last term ($\frac{24}{KC^2}$) gives the contribution of the inertia force to the total maximum force as a
proportion of the drag force. This inertia component is shown in the second– and third–last
columns of Tables S1 and S2.

When the inertia component ($\frac{24}{KC^2}$) contributes more than 25% of the drag force to the total
force, we consider that to be a potentially significant contribution. For example, when $F_I=F_D$,
the contribution to the maximum total force from the inertia force is $0.25F_D$. It should be
noted that this relationship is only valid for $\frac{F_I}{F_D} < 2$, and when $\frac{F_I}{F_D} > 2$, the maximum force is
pure inertia, meaning it is the dominant force (Dean and Dalrymple, 1991). The maximum
force is shown in the last column of Tables S1 and S2.

Table S1 (line 35) shows that only 18 out of 71 wave conditions in the flume have the
potential for the inertia force to be significant, and of those, only 6 had a $\frac{F_I}{F_D}$ ratio >2,
meaning that nearly all the wave conditions in the flume led to drag–dominated conditions.
Table S2 (line 42), confirms that only 1 out of 90 wave conditions in the field had the
potential for inertia to be significant. This condition corresponded to a very low velocity
(0.016 m/s), far from the reported transport threshold velocities. Thus, further
investigations were made for flume conditions (see below) but not field conditions.

The above calculations were applied to the dataset used to determine the probabilities of
rocking, flipping and transport in the flume (for loose and interlocked rubble), so that each
individual rubble piece's diameter could be used in place of the average diameter of
1.64 cm. Figure S5 (line 15) shows the relationship between bottom orbital velocity and the
$F_I/F_D$ ratio for every individual case in the flume. There is a general trend in which $F_I/F_D$
decreases as both velocity and the likelihood of movement increases. The dataset includes
7,593 rows and of these, 2,081 had the potential for inertia forces to be significant based
on the above calculations. However, in most (90%) of the 2,081 identified cases, there was
no movement of the rubble being tested (Figure S8 – line 26).

In 9.3% of cases (195 of 2,081), rocking movements only were recorded (Figure S6 – line
19). For these cases, the contribution of inertia force to the total force ranged from 25% to
100% or more of that contributed by the drag force. The highest velocity represented in
these cases was 0.2 m/s, though the large majority were much lower. Thus, at velocities
<0.2 m/s, there is the potential for inertia forces to contribute to causing rocking motions.
But, at a velocity of 0.2 m/s the contribution of inertia is still only 25% of the drag force (not
dominant). This is now indicated in the caption for the plot of velocity vs probability of
rocking (Figure 3a) in the revised manuscript (line 351) and as pasted below:

**"Figure 3: The probability of (a) rocking, (b) transport, and (c) flipping with increasing near-bed wave orbital**
**velocity for branched and unbranched rubble of four size categories (grey: 4-8 cm; green: 9-15 cm; light blue: 16-22**
**cm; dark blue 24-39 cm) on rubble and sand substrates. Note that at low velocities <0.2 m/s, we estimate there is the**
**potential for inertia forces to contribute to causing rocking motions; and at velocities <0.16 m/s, there is the potential**
**for inertia forces to contribute to causing transport and flipping."**

In any case, these instances of 'rocking' were considered as 'no movement' in the analysis
determining the probability of transport (see Table 1 of the manuscript) and thus have no
bearing on the 50% or 90% thresholds of transport reported.

Only in 0.9% of the cases where inertia forces were potentially significant (18 of 2,081), was
transport/flipping recorded (Figure S6 – line 19). For these cases, the average contribution
of inertia forces to the total force was 36% of the drag force (Figure S7 – line 23). The
highest velocity represented in these cases was 0.16 m/s (Figure S6 – line 19). This indicates
that at very low velocities <0.16 m/s, there is the potential for inertia forces to be
significant. However, this cut–off is well below the 50% and 90% thresholds of transport that
are reported in the paper, and at those velocities, i.e., ≥0.3 m/s, the inertia component
contributes as little as 0.1% and at most 4.9% to the total force, and the threshold of motion
conditions are thus drag dominated (Figure S8 – line 26).

The reviewer also states that "ideally, the probability of movement would be plotted against
a measure of the total force rather than velocity". We disagree that this is necessary, based
on the above calculations and justification, which are included in Supplementary Material
and text included in the revised manuscript. The total force is also directly dependent on the
length of the coral rubble pieces. The threshold of motion is not directly dependent on that
length, since, for a given velocity, doubling the length doubles both the force and the
resisting force. The observed thresholds are indirectly influenced by length through the
greater probability of a longer length piece having a shape that is more stable due to
curvature or branches. Furthermore, we feel a plot of movement against velocity is more
widely interpretable to a broad coral reef scientist audience, rather than only to those with a
knowledge of sediment transport and hydrodynamics, as flow speed is commonly measured
on reefs while forces are not. However, we do include tables showing the inertial force
component for each velocity (Tables S1 and S2), and plots of how the inertia force
component changes with velocity and movement (Figures S5 and S8, lines 15 and 26) in the
Supplementary Material.

To incorporate the above response into the revised manuscript we have included the
following sections, with line numbers corresponding to the manuscript with *no* track
changes.

Section 2.1 (Line 158)

"To determine whether scaling effects were necessary to compare velocity thresholds between flume and field
conditions, we derived a relationship for the contribution of the inertia force to the total maximum force as a
proportion of the drag force, for all wave conditions for each run. Total force depends on both the inertia force
and drag force components, and while the inertia component is dependent on velocity and wave period, the drag
component is solely dependant on velocity (Table S1). Thus, where conditions are determined to be drag
dominated, rubble movement depends primarily on velocity, and valid comparisons between flume and field can
be made despite their variance in wave period.

The inertia component and maximum force for each wave height and period combination in the flume, based on
an average coral diameter of 1.64 cm (range ~1-2 cm), are shown in Table S1. Only 19 out of 71 wave conditions in the flume have the potential for the inertia force to be significant, and of those, only 7 had a $\frac{F_I}{F_D}$

ratio >2, meaning that nearly all wave conditions in the flume led to drag-dominated conditions. Furthermore, the inertial component decreases as velocity increases (Figure S5), and inertial forces were negligible at the 50%

and 90% thresholds (see results for further explanation). The flume and field experiments are therefore comparable without scaling effects."

Section 2.2 (Line 211)

"The contribution of the inertia force to the total maximum force as a proportion of the drag force was estimated for each $H_s$ and $T_p$ combination used in the field analysis, based on an average coral diameter of 1.69 cm (range

~1-3 cm) (Table S2). Only 1 out of 90 wave conditions in the field had the potential for inertia to be significant, meaning that most conditions in the field were drag-dominated. Furthermore, this one condition corresponded to a very low velocity (0.016 m/s), far from the reported 50% and 90% transport threshold velocities."

Section 3.1.1 (Line 323)

"As well as calculating the inertia component for each wave height and period combination in the flume based on the average coral diameter (see 2.1 Methods), we also made these calculations for individual runs using the unique diameter of each piece. Of the cases identified as having the potential for inertia forces to be significant, 9.3%

(195 of 2,081) were runs where only rocking movements were recorded. The highest velocity represented in these cases was 0.2 m/s, though the large majority were much lower (Figure S6). Thus, at velocities <0.2 m/s, there is the potential for inertia forces to contribute to causing rocking motions. But, at a velocity of 0.2 m/s the contribution of inertia is still only 25% of the drag force (not dominant), and the threshold of rocking conditions in the flume, reported above, are drag dominated.

Transport or flipping occurred in only 0.9% of runs where we determined inertia forces to be potentially significant (18 of 2,081 runs) (Figure S7). For these cases, the average contribution of inertia forces to the total force was

36% of the drag force and the highest velocity represented in these cases was 0.16 m/s (Table S7). This indicates that at low velocities <0.16 m/s, there is the potential for inertia forces to be significant. However, this cut-off is well below the 50% and 90% thresholds of transport reported above, and at those velocities the inertia component contributes as little as 0.1% and at most 4.9% to the total force. The threshold of transport conditions in the flume are thus drag dominated."

We have also included description of the equations used to calculate inertia force in the

Supplementary Material on line 34 and pasted below.

**Table S1/S2 pre-amble:** Assuming the drag and inertia coefficients have the same magnitude, the ratio of the maximum inertia force to the maximum drag force is given by Dean and Dalrymple (1991) as:

$$(1)\quad \frac{F_I}{F_D} = \frac{\pi^2}{KC} = \alpha \qquad\qquad \text{where } KC = \frac{uT}{\varnothing}\ ; \text{KC = Keulegan-Carpenter number , u = maximum orbital}$$

wave velocity, T = wave period $\varnothing$ = rubble diameter

Hence $F_I = \alpha F_D$

The maximum total force is again given by Dean and Dalrymple (1991), noting that the drag and inertia forces are out of phase,

$$(2)\quad F_T = F_D + \frac{F_I^2}{4.F_D} \quad \text{FT = maximum total force, FD = drag force, FI =inertia force}$$

which can be written as $F_T = F_D + \frac{\alpha^2}{4}F_D$ or $F_T = F_D + \frac{24}{KC^2}F_D$

The last term $(\frac{24}{KC^2})$ gives the contribution of the inertia force to the total maximum force as a proportion of the drag force. We consider the inertia component to be potentially significant when it contributes more than 25% of the drag force to the total force. For example, when FI=FD, the contribution to the maximum total force from the inertia force is 0.25FD. It should be noted that this relationship is only valid for $\frac{F_I}{F_D} < 2$, and when $\frac{F_I}{F_D} > 2$, the maximum force is pure inertia (Dean and Dalrymple, 1991).

Specific comments

Abstract lines 10–15, and corresponding sections of the text. Comparisons rubble motion in
lab and field studies with respect to orbital velocities are flawed, due to the reasons outlined
above in my general comments. Also, rubble size is an important determinant of when
motion occurs, so I didn't understand why a single probability of motion at a single velocity
was reported.

Diagram – the statement that rubble is mobilized for orbital velocities greater than 0.4 m/s
is too simplistic since we know (and the results show) there is a strong dependence on
rubble size. There will also be a dependence on wave period for some rubble size classes
and wave conditions due to inertial force being the dominant force.

We thank the reviewer for these more detailed comments. As described above, the
comparisons between flume and field are made with respect to the 50% and 90% thresholds
of transport, and at those velocities, i.e., 50% at ≥0.3 m/s, the inertia component
contributes as little as 0.1% and at most 4.9% to the total force in the flume (Figure S8 – line
26), and conditions are thus drag dominated. At and above this same velocity threshold in
the field, the inertia component contributes on average 0.08% and a maximum of 1% to the
total force (Table S2). Thus, comparisons between flume and field are valid despite
differences in wave period. With respect to variation in rubble length, the graphical abstract
is a summary of the information provided in the paper, and the value given is the 90%
threshold averaged across substrates, morphologies and rubble lengths from 4-23 cm (and
diameters ~1–3 cm). This has now been highlighted in the footnote of the graphical abstract
which can be seen on line 38 of the revised manuscript and pasted below.

"50% chance of movement averaged across substrate, rubble morphology, rubble lengths 4-23 cm & diameters
~1-3 cm"

The figures in the manuscript provide the detail around rubble movement with respect to varying rubble lengths and morphologies, and we note in the discussion that "…interventions might be considered at lower mobilisation thresholds (e.g., 50% of 4-8 cm unbranched rubble predicted to move at 0.14 m/s in the field; Figure 6a) if a rubble bed is comprised predominantly of very small pieces" (Line 567). Additional supplementary tables of model predictions for figures can be included if additional detail is desired.

Page 6, line 5–10. A table of wave conditions in the flume (height, period, water depth) is needed. The description of how wave period was increased when waves started to break seems very arbitrary. Changing the wave period for the same wave height will alter the orbital velocity, orbital excursion, and acceleration.

Linear wave theory and the Soulsby model are less accurate once wave breaking occurs. Breaking wave conditions were thus avoided by changing the wave conditions to reduce the wave steepness. This alters the orbital velocity and forces, but has no bearing on the analysis, merely the order in which different conditions are achieved.

We have included a table of wave conditions used in the flume in the Supplementary Material (Table S1 – line 35). These show the average coral rubble diameter, significant wave height, period, water depth, and corresponding velocities, inertia force component and bottom orbital excursions for all wave conditions used in determining the relationship between velocity and movement.

P6, Line 17–20. Linear wave theory is generally used to estimate bottom orbital velocities, accelerations, excursions. The approach described here (Soulsby cosine approximation) is non-standard and I didn't understand why it was used in preference to linear wave theory.

The Soulsby cosine approximation is a one-step method to estimate bottom orbital velocity without solving the dispersion relationship, thus it was simpler to implement. A comparison of bottom orbital velocities estimated from linear wave theory compared to the Soulsby cosine approximation was conducted prior to submission of the original manuscript, for the wave conditions used in the flume. This relationship is shown in Figure S2 (line 4) of the Supplementary Material.

Velocities were found to be almost identical between methods, with an average change of 0.4 cm/s (55% <0.5 cm/s difference, 90% <1 cm/s difference) and a maximum change of 1.3 cm/s. A table of comparisons of these velocity estimations for each wave condition used in the flume is also included in Table S1 (line 35).

Based on the above relationship, we do not deem it necessary to re-run flume analyses using linear wave theory in place of the Soulsby Cosine Approximation.

P6, Line 19–20. The last statement on this page is very concerning: "Wave orbital velocities obtained in the flume were comparable to those measured in the field, hence scaling of the analyses was not required." As I explained above, the forces on the rubble are the relevant quantity that should be compared in the lab vs the field, and related to rubble motion. The velocities can be the same but if other important parameters are different (e.g., wave period) then direct comparisons of laboratory and field results will not be possible. Careful consideration of scaling is always required when relating lab flume experiments and the field situation.

Our response to the reviewer's general comment addresses this concern. As pointed out by
the reviewer, total force depends on both the inertia force component and drag force
component, and while the inertia component is dependent on velocity and wave period, the
drag component is only dependant on the velocity (see equations outlined above). Thus,
where conditions are determined to be drag dominated, rubble movement only depends on
the velocity. As outlined in our response above, our investigation found minor issues with
inertia becoming potentially significant at very low velocities in the flume, and some rubble
pieces (18 cases) transporting in those conditions, but the contribution of inertia forces to
total force in these cases was relatively small. Furthermore, the 50% threshold of ≥0.3 m/s
represent wave conditions that are drag dominated as in the field. Thus, despite differences
in the wave period between the field and flume, meaningful comparisons of these movement
thresholds can be made. Since inertial forces are negligible at the 50% and 90% thresholds,
the laboratory experiments do not have scale effects, since the coral rubble has the same
scale in the laboratory and field and the velocity and Reynolds numbers therefore also have
the same magnitudes.

P8. Line 10. Unclear that the shallow water approximation is valid here for computing
wavenumber k. There is readily available code available to calculate k from frequency and
depth using the general/complete linear wave theory dispersion relation.

We thank the reviewer for this comment and agree that in some circumstances where the
wave period is very short, the shallow water approximation should be avoided. We have now
solved the dispersion relation to calculate the wave number (k), and the near-bed orbital
velocities in the field have been updated using the new k values. This is outlined in the
manuscript as below:

"Pressure was converted to depth, and wave spectra for each 30-minute run were calculated between 0.0033-0.33
$H_z$ using the Welch method for computing power spectral densities from 3600 sample records, to obtain significant
wave height ($H_s$) and peak wave period ($T_p$). The near-bed wave orbital velocity (U) was then estimated for each
30-minute run using linear wave theory using Eq. (3).

(3) $U = \frac{H_s}{2 \sinh(kh)} \cdot \frac{2\pi}{T_p}$    where the wave number ($k$) was determined by solving Eq. (4)
(4) $\varpi^2 = gk \sinh(kh)$    where ω is the wave radian frequency ($2\pi/T_p$), $h$ is water depth, and g the
acceleration due to gravity."

A plot of the velocity as it appears in the original manuscript against the velocity using the
wave number as calculated in the revised manuscript is shown below, showing that
conditions where the shortest wave periods were observed (~4-5 s) result in the greatest
change.

[Figure]

Results in Section 3.2, as well as figures 4, 5 and 6 of the revised manuscript have been
updated based on the updated analysis. The analysis of 50% threshold was found to have
changed slightly from 0.34 m/s to 0.3 m/s (making the updated 50% threshold in the field
more like that estimated for the flume), and the 90% threshold from 0.55 m/s to 0.75 m/s.
The following is included in the results section regarding the 90% threshold:

"We note however that the 90% threshold for transport is above the range of velocities measured in the field and
should thus be considered cautiously compared to the 50% threshold. We do not report the 50% or 90% thresholds
for flipping in the field for the same reason."

P8. line 32. Unclear what is meant by peak wave orbital velocity here. Do you mean the
maximum 30–min significant wave height over the 3–day period? This is not truly the peak
wave orbital velocity, which would require going back to the original time series for each
burst.

The 'peak' wave orbital velocity in this manuscript is calculated based on the significant
wave height and the peak wave period from the wave spectrum, and we have selected the
fastest 'peak' wave orbital velocity per day for the regression with rubble movement. We
agree that the use of 'peak' in the referenced section of the manuscript may be confusing.

We have amended the text for clarity to now read:

"From the 30-minute runs across each 3-day period and site (144 each period and site), the fastest wave orbital
velocity (calculated from peak wave height and period) was selected for each day, to regress with observed rubble
movement on that day. A total of the 90 fastest wave orbital velocities were thus used in the analyses that included
all three days (1 velocity per day x 3 days x 15 sites x 2 seasons), and 30 were used in the analyses that included
the first day only (1 velocity for each 'day 1' x 15 sites x 2 seasons)."

 **P9. Results. Wave periods need to be reported and used appropriately in the analysis!**

Tables of wave conditions in the flume and field, including average coral rubble diameter,
wave height, period and water depth are now provided in the Supplementary Material as
Tables S1 and S2. The additional text in the revised manuscript relating to this is pasted
above from line 198. There is also a preamble to Table S1/S2 (line 34) explaining the
equations used to calculate the inertia force component, as pasted below:
* * *
**Table S1/S2 pre-amble:** Assuming the drag and inertia coefficients have the same magnitude, the ratio of the maximum inertia force to the maximum drag force is given by Dean and Dalrymple (1991) as:

(1) $\frac{F_I}{F_D} = \frac{\pi^2}{KC} = \alpha$          where $KC = \frac{uT}{\varnothing}$ ; KC = Keulegan-Carpenter number , u = maximum orbital wave velocity, T = wave period $\varnothing$ = rubble diameter

Hence $F_I = \alpha F_D$

The maximum total force is again given by Dean and Dalrymple (1991), noting that the drag and inertia forces are out of phase, (2) $F_T = F_D + \frac{F_I^2}{4.F_D}$   FT = maximum total force, FD = drag force, FI =inertia force which can be written as $F_T = F_D + \frac{\alpha^2}{4} F_D$ or $F_T = F_D + \frac{24}{KC^2} F_D$

The last term ($\frac{24}{KC^2}$) gives the contribution of the inertia force to the total maximum force as a proportion of the drag force. We consider the inertia component to be potentially significant when it contributes more than 25% of the drag force to the total force. For example, when FI=FD, the contribution to the maximum total force from the inertia force is 0.25FD. It should be noted that this relationship is only valid for $\frac{F_I}{F_D} < 2$, and when $\frac{F_I}{F_D} > 2$, the maximum force is pure inertia (Dean and Dalrymple, 1991).
* * *

 **Fig 4. Bottom orbital excursion and accelerations should be shown also, for the reasons**
**outlined in my General Comments**

We have included a table of wave conditions used in the flume (Table S1 – line 35) and the
field (Table S2 – line 42) in the Supplementary Material of the revised manuscript. These
show the average coral rubble diameter, wave height, period, water depth, and
corresponding velocities, inertial force component and bottom orbital excursions for all
wave conditions used in determining the relationship between velocity and movement. As
one of the key goals of this work is to inform management around reef restoration, we feel
that the presentation of the results (figures) as a function of velocity is more broadly
comprehendible to managers than are forces.